METHODS AND RESOURCES

# Protein nanobarcodes enable single-step multiplexed fluorescence imaging

**Daniëlle de Jong-Bolm**[1]☯*, **Mohsen Sadeghi**[2]☯*, **Cristian A. Bogaciu**[1], **Guobin Bao**[3], **Gabriele Klaehn**[1], **Merle Hoff**[1], **Lucas Mittelmeier**[1], **F. Buket Basmanav**[1,4]¤, **Felipe Opazo**[1,5,6], **Frank Noé**[2,7,8,9], **Silvio O. Rizzoli**[1,6]*

1 Department of Neuro- and Sensory physiology, University of Göttingen Medical Center, Cluster of Excellence "Multiscale Bioimaging: from Molecular Machines to Networks of Excitable Cells" (MBExC), Göttingen, Germany, 2 Department of Mathematics and Computer Science, Free University of Berlin, Berlin, Germany, 3 Institute of Pharmacology and Toxicology, University Medical Center, Georg-August-University, Göttingen, Germany, 4 Campus Laboratory for Advanced Imaging, Microscopy and Spectroscopy, University of Göttingen, Göttingen, Germany, 5 Center for Biostructural Imaging of Neurodegeneration (BIN), University of Göttingen Medical Center, Göttingen, Germany, 6 NanoTag Biotechnologies GmbH, Göttingen, Germany, 7 Department of Physics, Free University of Technology, Berlin, Germany, 8 Department of Chemistry, Rice University, Houston, Texas, United States of America, 9 Microsoft Research AI4Science, Berlin, Germany

☯ These authors contributed equally to this work.
¤ Current address: Institute of Human Genetics, University of Bonn, Medical Faculty & University Hospital Bonn, Bonn, Germany
* danielle.dejong@med.uni-goettingen.de (DdJ-B); mohsen.sadeghi@fu-berlin.de (MS); srizzol@gwdg.de (SOR)

## Abstract

Multiplexed cellular imaging typically relies on the sequential application of detection probes, as antibodies or DNA barcodes, which is complex and time-consuming. To address this, we developed here protein nanobarcodes, composed of combinations of epitopes recognized by specific sets of nanobodies. The nanobarcodes are read in a single imaging step, relying on nanobodies conjugated to distinct fluorophores, which enables a precise analysis of large numbers of protein combinations. Fluorescence images from nanobarcodes were used as input images for a deep neural network, which was able to identify proteins with high precision. We thus present an efficient and straightforward protein identification method, which is applicable to relatively complex biological assays. We demonstrate this by a multicell competition assay, in which we successfully used our nanobarcoded proteins together with neurexin and neuroligin isoforms, thereby testing the preferred binding combinations of multiple isoforms, in parallel.

## Introduction

Fluorescence imaging is one of the most powerful tools for cellular investigations, but its potential to reveal multiple targets has been rarely fulfilled, due to difficulties in labeling many molecules simultaneously or in separating multiple fluorophores spectrally [1]. One potential solution has been the introduction of multiplexing by sequential labeling, in which reagents carrying the same fluorophore are added and removed sequentially. This can be achieved by

**Data Availability Statement:** The software package Deep-Nanobarcode is publicly available as an open source software under the terms of the MIT license via the repository https://github.com/

noegroup/deep_nanobarcode. Optimized network hyperparameters, weights of the trained networks, and training datasets are all freely available for download from the ftp server ftp://ftp.mi.fu-berlin. de/pub/cmb-data/deep_nanobarcode. The software package additionally includes functionality for automatically downloading all the needed data. Images used for training and testing the deep network are available in the original format (LSM) from a Refubium repository hosted by the Free University of Berlin and accessible via the link http://dx.doi.org/10.17169/refubium-39512. The rest of the data presented in this study are available from a secondary Refubium repository available via the link http://dx.doi.org/10.17169/refubium-40101.

**Funding:** European's Union Horizon 2020 Horizon research and innovation program under grant agreement No 964016 (FET-OPEN Call 2020, IMAGEOMICS project). https://cordis.europa.eu/project/id/964016 M.S. and F.N. received financial support from Deutsche Forschungsgemeinschaft (DFG) through grants CRC 958/Project A04 (https://www.sfb958.de/de/index.html) and CRC 1114 (http://www.mi.fu-berlin.de/en/sfb1114/). F. N. was additionally supported by European Research Commission grant ERC CoG 772230 (https://cordis.europa.eu/project/id/772230), Bundesministerium für Bildung und Forschung (BMBF) grant 031L0195 "AutoXRayCell" (https://www.bmbf.de/bmbf/de/home/home_node.html), The Berlin Mathematics Center MATH+, projects AA1-6, AA1-10 (https://mathplus.de/) and the Berlin Institute for Foundations in Learning and Data (BIFOLD, https://bifold.berlin/de/). F.B.B. was supported by the Deutsche Forschungsgemeinschaft (DFG) through Cluster of Excellence Nanoscale Microscopy and Molecular Physiology of the Brain (CNMPB, http://www.cnmpb.de/) and by the Campus Laboratory for Advanced Imaging, Microscopy and Spectroscopy (AIMS, https://www.uni-goettingen.de/de/532762.html). Additional support comes from the DFG under Germany's Excellence Strategy (EXC 2067/1-390729940) and SFB1286 projects A03 and Z04 (https://www.sfb1286.de/). The funders had no role in study design, data collection and analysis, decision to publish, or preparation of the manuscript.

**Competing interests:** S.O.R. and F.O. are shareholders of NanoTag Biotechnologies GmbH. All other authors declare no potential conflict of interest.

**Abbreviations:** EGF, epidermal growth factor; FACS, fluorescence-activated cell sorting; HEK293, human embryonic kidney 293; kPCA, kernel

fluorophore bleaching (for example, in toponome mapping [2]), by antibody removal using harsh buffers, or by probe removal by extensive wash-offs (for example, maS³TORM [3] or DNA-PAINT [4]). While these approaches have been used to investigate samples from cancer cells to synapses, they involve long-lasting and challenging experiments and typically result in vast amount of data. Deep learning, driven by artificial neural networks, is a versatile solution for processing large datasets, which enables efficient quantitative analysis and extraction of features [5]. Despite the reduction of tedious manual analyses of large datasets, deep learning does not remove the more common challenges of multiplexing experiments, as long time periods necessary for imaging, and the increased chance of sample or experiment failure during multistep operations.

A simpler and more straightforward solution for multiplexing is presented here. We started from the idea that every microscope has a handful ($n$) of spectrally distinguishable channels, with which $n$ specific labels should be differentiated relatively easily. The number of possible combinations of labels is substantially higher than $n$, since each label can be present or absent ("on/off" signals), which leads, in theory, to $2^n$ combinations, as in a conventional barcode. As the "all labels absent" combination is useless for practical purposes, the actual number of targets that could be differentiated becomes $2^n-1$. Therefore, this barcoding approach could be used to strongly enhance the number of targets that can be analyzed simultaneously using a limited number of channels. So far, it has been used for cell identification by fluorescence-activated cell sorting (FACS; [6]), using antibody detection, but could not be yet introduced in the domain of conventional microscopy. Imaging the different label combinations using antibodies is almost impossible, due to problems with steric hindrance caused by the large antibody size, label clustering induced by the dual binding capacity of the antibodies, and limited epitope availability due to poor penetration into the cells [7,8]. Therefore, we relied here on epitope recognition by nanobodies (single-domain camelid antibodies), which are monovalent and substantially smaller than antibodies [9,10]. As a first step, we engineered proteins that contain a combination of 5 genetically encoded epitopes that are recognizable by nanobodies. The recognized combinations were termed "nanobarcodes." Second, we established a deep network, which was used for the automatic identification of nanobarcoded proteins. In essence, this deep network is a composition of simple nonlinear functions with adjustable parameters forming an extremely flexible, yet trainable map. Our artificial neural network functions as a pixel-wise classifier, which reads and decodes nanobarcodes from single pixels of fluorescence images, decides which protein is most likely represented by a particular pixel, and assigns a predefined false color representing a specific protein. As a result, the input image is transformed into a protein identification map. Finally, we provide this open source solution for reading and translating nanobarcode images into single proteins maps, including the necessary software and initial datasets, for training and use in other laboratories.

## Results

### Nanobody-based identification of barcoded proteins using simple immunocytochemistry

We engineered our barcoded proteins containing up to 5 nanobody epitopes as follows. First, a reference epitope was added to all our barcodes, in the form of the ALFA-tag [11]. This tag forms a small and stable α-helix, and its functionality is irrespective of its position on the target protein [11], thereby enabling us to detect every barcode, irrespective of what other epitopes are present. The other 4 epitopes were present only in subsets of all barcodes: mCherry(Y71L) and GFP(Y66L), both mutated to generate nonfluorescent variants [12] and 2 different short sequences found at the C-terminus of human α-synuclein [13] (termed here syn87 and syn2).

Principal Component Analysis; NLS, nuclear localization signal; t-SNE, t-distributed Stochastic Neighbor Embedding; VAMP, vesicle-associated membrane protein.

These 4 epitopes were engineered, in different combinations, into the sequences of different proteins, and were then revealed using the respective fluorescently labeled nanobodies (NbRFP, NbEGFP, NbSyn87, NbSyn2). We call these nanobody-revealed barcodes nanobarcodes. As designed, all epitopes were easily detected in immunocytochemistry (Fig 1). We implemented the barcodes in 15 different proteins ($2^4-1$), according to the schemes shown in Fig 1A–1D. We targeted proteins mostly from the secretory pathway (Fig 1E), such as vesicle-associated membrane proteins (VAMPs) and Syntaxins. A schematic topology of all protein constructs is provided in S1 Fig.

## Validation of nanobarcoded protein organization and function

As illustrated in Fig 1F, the nanobarcodes can be easily differentiated by the human observer. Correct expression of the target proteins (Fig 1C) used in our barcoded constructs was validated as follows. Instead of nonfluorescent constructs, constructs with fluorescent mCherry and GFP epitopes were used, enabling a direct visualization of the proteins. Target proteins were visualized with immunocytochemistry, relying on antibodies, using wide-field microscopy in a common cell line, HEK cells. In this way, the expression patterns of barcoded and endogenous target proteins are revealed and compared (S2 Fig), providing one layer of validation for all constructs.

Further validation of all constructs was achieved by the successful visualization of each of our barcodes using 4 fluorescent nanobodies (S3 Fig). Neither the genetically induced loss of fluorescence of the EGFP and mCherry epitopes, nor the number of epitopes per se, seem to hinder the nanobodies in binding to their respective epitopes (S4 Fig).

We then proceeded to another layer of validation, this time aiming to understand whether the barcoding affected the location and/or function of the proteins. However, some of our barcoded proteins do not actually have a cellular function. This subset includes cytosolic GFP, the nuclear localization signal (NLS), the ER-retention signal (KDEL sequence), and a mitochondria localization sequence (TOM70). To determine whether the epitopes alter the behavior of these proteins, we analyzed their colocalization to the compartments in which they should be present, relying on 2-color microscopy experiments (S5 Fig). We also added GalNacT to this experiment, because its localization in the Golgi apparatus is essential for its function [14], and because functional assays for this protein are not easily implemented by microscopy experiments.

All other nanobarcoded proteins are involved in membrane trafficking in the cell, which implies that they can be readily tested by classical assays designed to test receptor and cargo trafficking. We chose 2 such assays, which were performed in parallel. First, we used a transferrin endocytosis and recycling assay. The protein transferrin is involved in iron metabolism in all mammalian cells, and is readily endocytosed, upon binding to its receptor. Transferrin is then recycled and released from the cells, within a time frame of a few tens of minutes [15]. This enables the microscopy investigation of both transferrin uptake during pulsing with fluorescently conjugated transferrin, as a measure of endocytosis potential, and transferrin loss after a chase, as a measure of recycling and exocytosis. Second, we relied on the endocytosis of the epidermal growth factor (EGF) receptor. The addition of fluorescently conjugated EGF onto the cells results in abundant ligand-mediated endocytosis of the receptors, which are not recycled, but proceed slowly to the lysosomal compartment, where they are later degraded [16]. Therefore, no substantial loss of EGF fluorescence is expected upon a chase of a few tens of minutes, offering a different readout to transferrin.

We performed both of these assays for the endosomal membrane organizer Rab5a, for Lifeact (whose binding to actin should lead to a small, but measurable enhancement of actin

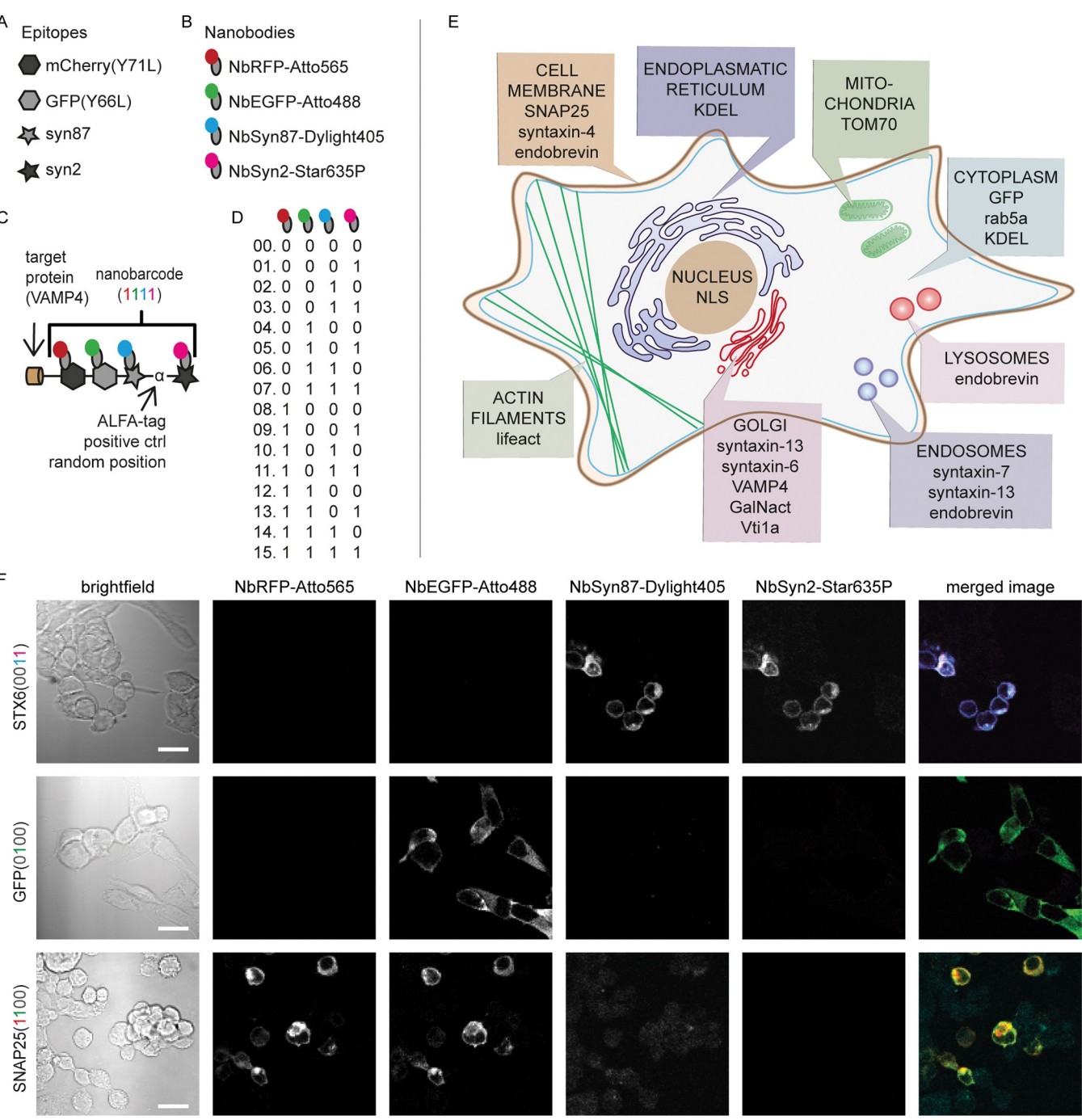

**Fig 1. Design of protein constructs with nanobarcodes using 4 nanobody epitopes.** (**A**, **B**) Scheme of the 4 nanobarcode epitopes (**A**) and the fluorescent nanobodies used for recognizing them (**B**). (B) NbRFP-Atto565 in red, NbEGFP-Atto488 in green, NbSyn87-Dylight405 in cyan, NbSyn2-Star635P in magenta. (**C**) Design of the protein construct VAMP4(1111). Each protein construct contains a target protein (the protein to identify) and a barcode. In this example, the target protein is VAMP4, and its barcode contains the following nonfluorescent epitopes: mCherry (Y71L), GFP (Y66L), syn87, and syn2. The ALFA-tag [10] is present for testing purposes. See S1 Fig for further sequence information. Barcode epitopes recognized by fluorescent nanobodies shown as "ones" in pseudocolors that correspond to the fluorophores used. (**D**) Nanobarcodes, 15 in total, resulting from a binary combination of 4 nanobarcode-epitopes. Epitopes from left to right: mCherry(Y71L), GFP(Y66L), syn87 and syn2. The nanobody scheme is the same as in (**B**). (**E**) The expected cellular protein distribution for the proteins used, according to the literature. (**F**) Nanobarcode-based identification of the proteins STX6(0011), GFP(0100), and SNAP25(1100). The pseudocolors for merged images correspond to the fluorescence channels of the nanobodies: NbRFP-Atto565 in red, NbGFP-Atto488 in green, NbSyn87-Dylight405 in cyan, and NbSyn2-Star635P in magenta. Scale bar: 20 μm.

dynamics [17]), and 7 SNARE molecules involved in fusion events in the membrane trafficking pathway: endobrevin, syntaxins 4, 6, 7, and 13, Vti1a, and VAMP4. The expected result is that the overexpression of these proteins will not affect the transferrin and EGF dynamics negatively but would rather lead to small enhancements of their uptake (and possibly release as well, for transferrin). The changes induced by the expression of barcoding proteins can only reach a moderate level, since the respective trafficking pathways remain limited by the abundance of many other proteins, which are not overexpressed. We obtained this result for all proteins. S6 Fig presents an overall view of the results, indicating the transferrin and EGF dynamics in all experiments, combined. S7–S15 Figs show the results for every individual protein, comparing the transferrin and EGF signals to the levels of overexpression of the respective proteins. Overall, these experiments indicate that these components of the membrane trafficking machinery are not negatively affected by our tagging procedure.

One additional SNARE molecule, SNAP25, is more difficult to test in such experiments, since it only functions in synapses, where its abundance is already extraordinary [18], so that overexpression is not expected to lead to changes in synaptic processes (just as lowering SNAP25 levels in heterozygous SNAP25+/− mice leads to very minor phenotypes [19]). To validate the behavior of SNAP25, we therefore relied on a super-resolution imaging assay, in which we tested its localization, in comparison to endogenous SNAP25, in a neuroblastoma cell line (PC12). The results, shown in S16 Fig, indicate that our epitope tagging does not affect SNAP25 localization. A quantification of localization results, also including the work relating to the proteins lacking a cellular function, is shown in S17 Fig.

## Deep learning–based identification of protein nanobarcodes

While the identification of protein nanobarcodes, which fluoresce as a combination of their tags, is possible with the human eye (Figs 1F and S3), it is still a statistical inference task and, accordingly, is more suited for automated machine learning algorithms. The task can be formulated in simple terms as to determine the probability of the observed protein belonging to one of $2^n$-1 categories, given the registered intensities in all the microscope channels. This inference is to be done for each pixel in the image to translate the spectral block into a bitmap representation with proteins highlighted in false colors.

We used deep learning for the nontrivial classification task and developed the Deep-Nanobarcode software package (https://github.com/noegroup/deep_nanobarcode). Deep-Nanobarcode is a Python package developed using PyTorch machine learning framework and deploys a deep neural network trained to map the combined fluorescence output of the nanobarcode sequences to the identity of the respective labeled proteins (Figs 2 and S18).

Our developed deep network is essentially a pixel-wise classifier, has about 620k trainable parameters, and is trained in a supervised manner (Figs 2A and S18, Methods section "Deep neural network for nanobarcode identification"). The training dataset is gathered from confocal images of single-transfect samples using a machine-learned thresholding scheme (see Methods section "Data pipeline for training and evaluation of the deep network" and S3 Table). These data are split for training, validation, and testing (Methods section "Training and testing the deep network").

Additionally, we have provided the possibility of invoking another level of machine learning at inference time when using whole images as input. This is achieved via a trainable contrast-modifier acting in tandem with the deep network, which is trained in a self-supervised manner (Methods section "Training and testing the deep network"). We found that training the contrast-modifier with small number of steps (between 10 and 100) helps with enhancing the sparsity in the prediction, i.e., less noisy predictions in the image backgrounds. Essentially, the

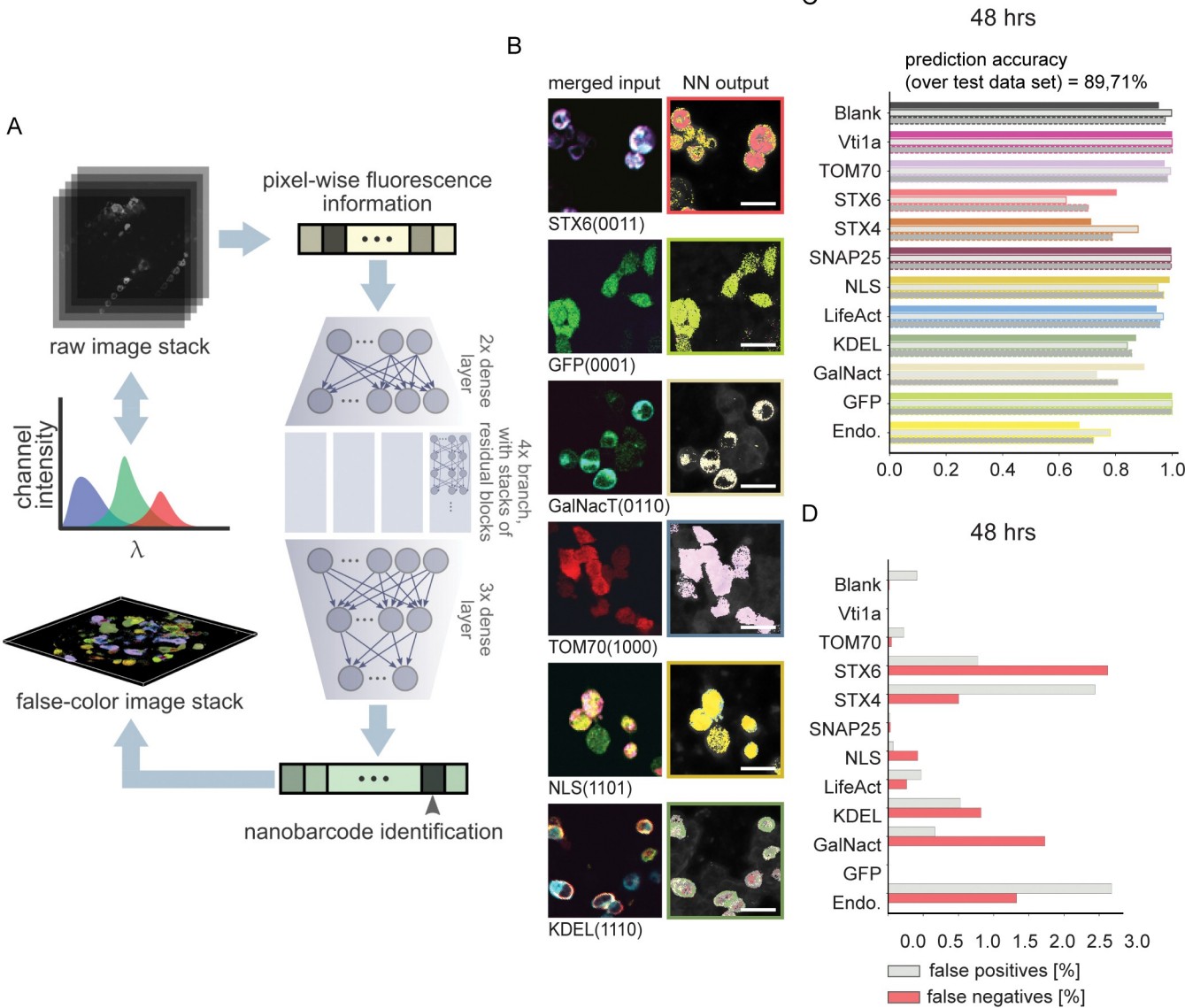

**Fig 2. Neural network–based identification of nanobarcode proteins.** (**A**) Schematic of the neural network used for identification of nanobarcodes from pixel-wise fluorescence information. Brightness values across all emission channels are fed to the network as input, which, in turn, has been trained to predict the probability of this information pertaining to a specific nanobarcode, or a blank pixel. The trained network can readily be applied to full micrographs as well as stacks of images to produce false color outputs illustrating spatial distribution of proteins (further details in S18 Fig). (**B**) Example images of HEK293 cells transfected with specific nanobarcodes. To account for all possible emission features (including bleed-through), we acquired 11 frames for each area, consisting of the following: 405 nm excitation, with emission windows in blue, green, red, deep red; 488 nm excitation, with emission windows in green, red, deep red; 561 nm excitation, with emission windows in red and deep red; 633 nm excitation, with an emission window in deep red; brightfield. The panels in the left column show an overlay of the 4 brightest frames: 405 nm excitation, blue emission (in cyan); 488 nm excitation, green emission (in green); 561 nm excitation, red emission (in red); 633 nm excitation, deep red emission (in magenta). False color neural network output images are shown in the right column of (**A**). (**C**) Prediction accuracy of the neural network over a hold-out test dataset. For each protein, bars represent the precision (top), recall (middle), and F1-score (bottom). (**D**) False positive and false negative protein identifications (as percentage of all false predictions). For further details about the experimental procedures, imaging settings and neural network analysis, see the Methods section. For practical implementation purposes, we concentrated here on a subset of the labeled proteins, which were also used for the Nrxn/Nlgn experiments in Fig 4. Scale bars: 20 μm. The data underlying this Figure are available as file "Fig 2_CD.xlsx" from http://dx.doi.org/10.17169/refubium-40101.

contrast-modifier's target of reducing the entropy in the network output helps remove spurious detection of nanobarcodes with weak or noisy input signals. But, of course, its training procedure is agnostic to the correct nanobarcode to be picked, and no new information would be gained with more training steps.

With the data being processed on the fly through our data augmentation protocol (Methods section "Training and testing the deep network"), and utilizing a GeForce RTX 3090 graphics card with 24 GB of graphics memory, fully training the network on our dataset takes up to 2 hours in each case. After training the network, and utilizing the same hardware, the inference takes about 15 seconds for each $512 \times 512$ pixel image, when an additional 50 iterations of self-supervised contrast adaptation is performed. While this deep learning framework can readily be fine-tuned or retrained on new imaging data, we provide all the weights of the network trained for the cases discussed here. The Deep-Nanobarcode software can thus be applied out of the box to new confocal images containing the same nanobarcodes described here, without the need for retraining.

## Evaluation of the performance and reliability of the deep network

After training the network, we analyzed its performance on (i) hold-out test sets and (ii) full images of single-transfected samples containing known nanobarcodes. The metrics we have used for evaluation of network performance are the percentage of false positive and negatives, accuracy, recall, and F1-score (Methods section "Training and testing the deep network"). Analysis on hold-out test sets, to which the network has not been exposed during any stage of training and validation, revealed a prediction accuracy of at least 80% for all the cases (Fig 2C and 2D). The analysis on full images resulted in a relatively high accuracy, considering the strong criterion of pixel-wise true identification (Figs 2B–2D and S20). Generally, optimal precision was achieved when the network was trained and tested on samples with similar expression and transfection time windows (S20 Fig). For these cases, we calculated a mean pixel-wise precision of 70% with 95% confidence interval of (63%, 77%).

For this analysis task, the use of a neural network was inevitable, as shown in Fig 3. We exhausted the possibility of using shallow machine learning algorithms for the analysis. We fed the data gathered for known proteins, as pixel-wise intensities in 10 channels, into 4 well-known dimensionality reduction algorithms, namely, the Isomap Embedding [20], kernel Principal Component Analysis (kPCA) [21], t-distributed Stochastic Neighbor Embedding (t-SNE) [22], and Spectral Embedding methods ([23]; Fig 3A and 3B). While we achieved some successful separation with more obvious cases, such as GFP, none of these methods are able to partition the whole dataset into a meaningful set of clusters.

We further performed an ablation study to establish the sensitivity and reliability of predictions. In a series of experiments with the deep network, we removed proteins one by one from the training data, fully trained the network on the remainder of samples, and measured its performance (Fig 3D). Generally, reducing the number of target classes in training the network improves its performance, as the task of mapping input vectors to the classes becomes easier. Nevertheless, this inevitable bump in performance is not uniform for all the target proteins, and, not surprisingly, removal of proteins that have the lowest prediction scores results in higher increase in performance (Fig 3D). This finding implies that the respective low prediction scores are inherent to the data gathered for the corresponding nanobarcodes and not a shortcoming of the deep network.

## Proof-of-principle application of the deep network for neuronal cell biology

After ensuring that the network could identify proteins with satisfactory precision, we set out to apply it to samples in which the combinations of transfected cells were unknown. We

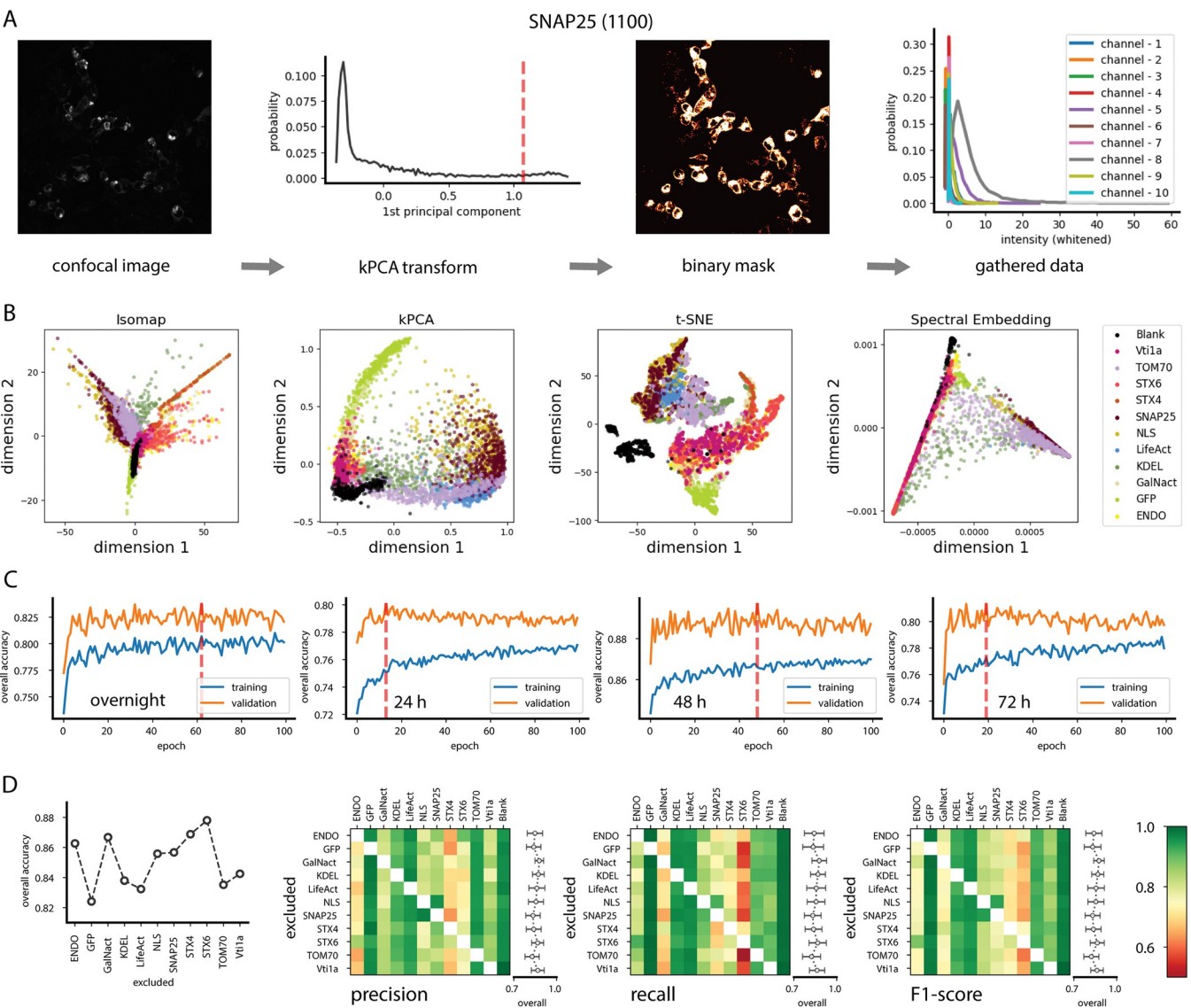

**Fig 3. Training and testing of the deep network.** (**A**) Pipeline through which data are prepared for training and testing the deep network for SNAP25 from 48-hour protocol as an example. Ten-dimensional vectors containing pixel-wise intensities across all channels are mapped along one dimension using kPCA transform. A relative threshold on the principal component separates foreground from background and results in a binary mask, based on which data can be gathered from points than contain proteins in the confocal image. (**B**) The result of Isomap, kPCA, t-SNE, and Sepctral Embedding "shallow-learning" methods for dimensionality reduction applied directly to the data gathered according to the pipeline explained in (**A**). (**C**) Training and validation accuracies averaged over all proteins in the dataset, sampled in each training epoch. Red dashed line shows the early stopping used based on the monitored validation accuracy. (**D**) Results of the ablation study, in which in each case one protein is removed from the training dataset and the performance of the deep network is evaluated based on the given metrics after training and validation procedure is performed. The data underlying this Figure are available as file "Fig 3_ABCD. xlsx" from http://dx.doi.org/10.17169/refubium-40101.

considered the prediction precision of more than 80% on test data and mean prediction precision of 70% on image data to suffice for the purpose of reliably localizing molecules in the biological task. The way we measured precision in these examples is very strict, since it includes all the accumulated effects of (i) expression of nanobarcodes in imaged cells, (ii) imaging conditions, and (iii) uncertainty in deep network predictions, into one final score. Therefore, we consider the overall prediction precision to be satisfactory.

To apply this analysis to a relevant biological problem, we turned to a set of cell adhesion molecules that are essential in neuronal cell biology: neurexins (Nrxns1-3), found in the pre-synapse, and neuroligins (Nlgns1-4), expressed in the post-synapse. These molecules are essential for synaptic regulation. These molecules bind to each other and to other partners in neuronal cells, inducing the formation of synapses. Their mutation and/or deletion can lead to the loss of synapses [24]. Both Nrxns and Nlgns can be used in vitro, in experiments in which individual cells express some of these molecules, enabling then to form "synapses" between them [24]. In such experiments, any combination of Nrxns and Nlgns could result in synapse formation. This does not take place in the brain, where specific interactions tend to take place, possibly due to further complexity in the behavior of these molecules. They contain glycosylation domains [25], a posttranslational modification that makes it very likely that these molecules are endocytosed and recycled, in order to repair damage to the glycosylation in a re-glycosylation mechanism that has been described for more than 2 decades in cancer cell cultures (e.g., [26] but has only recently been related to the synapse [27]). This implies that these molecules may have complex behaviors at the cell surface, including detailed membrane trafficking, as discussed already for both Nrxns and Nlgns, which will modify their capacity to interact with each other [28,29]. Moreover, binding between Nrxns and Nlgns is dependent on the alternative splicing of these molecules, resulting in a complex pattern of interactions [30].

Overall, Nrxn/Nlgn binding is subject to detailed and poorly known regulation, with the interaction of specific partners being affected by neuronal plasticity and by local conditions, and also by their membrane trafficking behavior. Interactions between the 2 sets of molecules are typically investigated by introducing single splicing variants into cells, followed by a one-by-one comparison of binding properties and/or interactions between Nrxn/Nlgn pairs expressed on different cells that are combined in vitro [31–35]. This type of analysis can pinpoint the interactions with the highest affinity, but they do not necessarily recapitulate the in vivo situation. Ideally, cells carrying different Nrxns and Nlgns should be exposed to each other simultaneously, in a multicell competition, to enable individual cells to test different potential partners, as in living tissues.

We therefore applied the nanobarcoding tools to this problem (Figs 4 and S20). We coexpressed different Nrxns and Nlgns with specific barcoded proteins (S20 Fig), and we then developed a cell-seeding assay that allows us to map all of the respective Nrxn/Nlgn interactions (Fig 4). We applied this assay to 4 β-Nrxns and 7 Nlgn isoforms: Nrxn-1ß (SS#4(+)), Nrxn-1 (SS#4(−)), Nrxn-2ß (SS#4(−)), Nrxn-3ß (SS#4(−)), Nlgn1(−), Nlgn1 (SS#B), Nlgn1 (SS#AB), Nlgn2 (−), Nlgn2 (SS#A), Nlgn3(WT), Nlgn4 (WT) (see also S21 Fig).

From the total number of cell contacts made by each Nrxn- or Nlgn-positive cell (using the nanobarcodes as reference), we calculated the percentage of specific Nrxn/Nlgn pairs (Fig 4B–4D). We found that some specific combinations are substantially more likely than others (Fig 4D). Like the Nrxn2ß (SS#4(+))/Nlgn1(−), we regularly identified Nrxn1ß (SS#4(+))//Nlgn3 (WT) pairs, which is surprising, since Nlgn3 is thought to have a lower affinity for Nrxns than the Nlgn1 and Nlgn2 isoforms [35]. In addition, 3 other Nrxn/Nlgn pairs were observed regularly: Nrxn1ß (SS#4(+))/Nlgn1(−), Nrxn2ß (SS#4(−))/Nlgn2 (SS#A) and Nrxn3ß (SS#4(−))/Nlgn2 (SS#A), which are compatible with the previous literature, albeit none are known to be of particularly high affinity. This implies that such an assay should be used for testing further the Nrxn/Nlgn interactions, especially as it is able to take into account not only the molecular binding but also the further dynamics that are induced by binding, such as molecular endocytosis and trafficking [29].

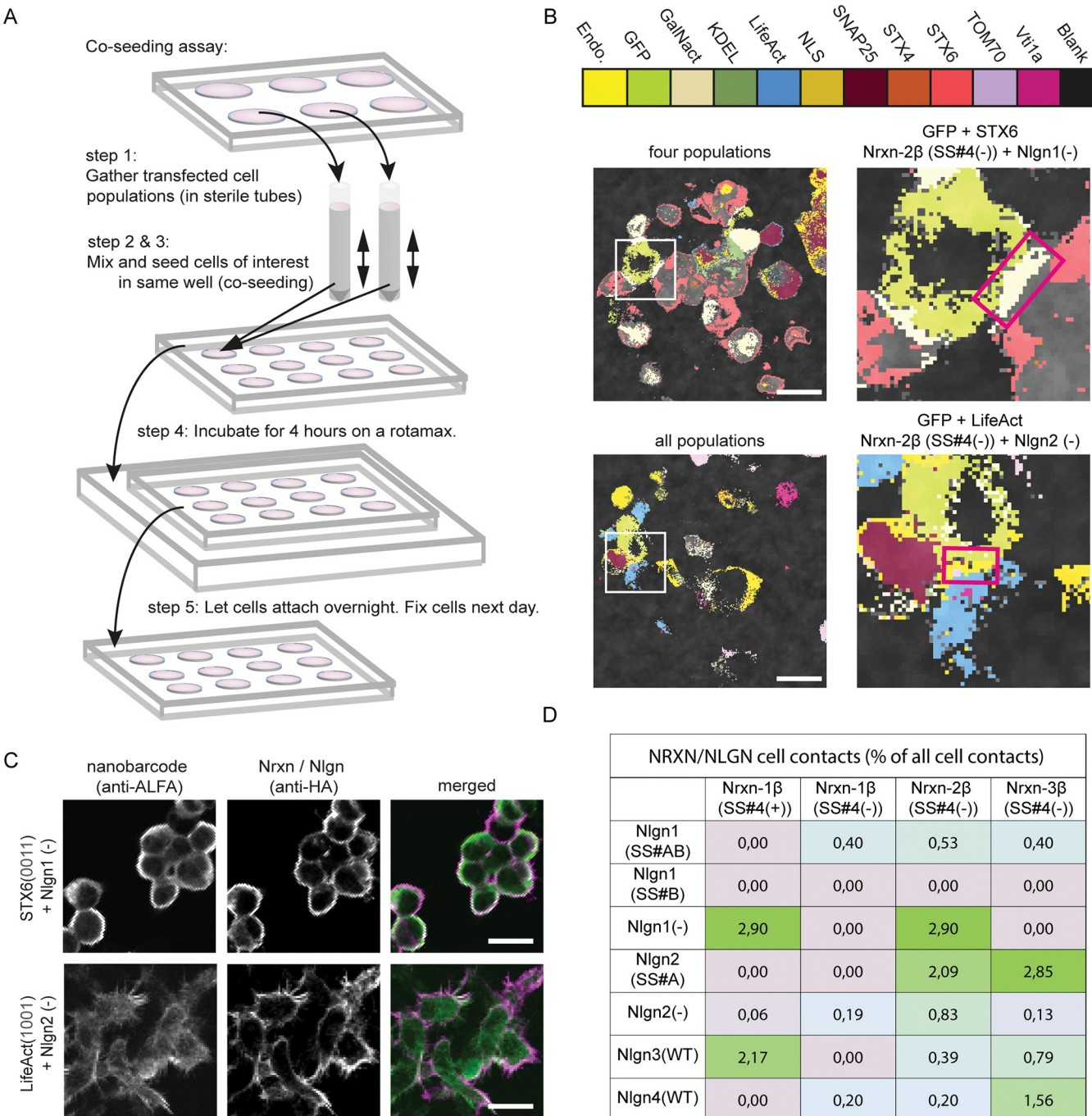

**Fig 4. Multiplex identification of proteins using a neural network–based spectral analysis.** (**A**) Experimental design of a co-seeding assay including 11 different cell types, labeled with specific nanobarcodes (see Methods section for details). (**B**) Example of an Nrxn-2ß (SS#4(+))/Nlgn-1 (SS#AB) and an Nrxn-2ß (SS#4(−))/Nlgn-2 (−) pair (red boxes depict typical cell contacts). (**C**) Overlay of cells containing nanobarcode proteins and Nrxn- or Nlgn-positive cells. Nanobarcode proteins are shown in green (anti-ALFA-Atto488). Nrxn or Nlgn isoforms are shown in magenta (anti-HA and anti-goat-Cy3). See S21 Fig for example images of all proteins. Scale bars: 20 μm. (**D**) Interaction preferences of Nrxn/Nlgn isoforms. A total of 4,569 cell contacts, 147 images, 4 independent co-seeding experiments. The Nrxn/Nlgn codes, such as SS#4(+) refer to the respective splicing sites of the proteins, according to the literature (e.g., [24]). The data underlying this Figure can be found in the S1 Data file, Sheet "Fig 4D", available from http://dx.doi.org/10.17169/refubium-40101.

## Discussion

We conclude that the nanobarcoding technology is feasible in conventional microscopy assays. We would like to point out that the error measured for the prediction precision on images (S20 Fig) originates from a mixture of machine learning performance and the whole pipeline of expression, immunostaining, and imaging the nanobarcodes. Considering this compound effect, our results appear to be highly effective in identifying and localizing proteins in crowded biological samples, using simple, conventional imaging tools.

One limitation of applying this method in conventional imaging is that single pixels could reflect the emission of many proteins, and our analysis will only indicate the most common one present in the respective pixel. In our present work, single pixels typically reflect only one type of nanobarcode, since the tagged molecules are expressed in different compartments, with limited overlap, whenever they are combined. To avoid this problem, if the nanobarcodes find themselves in the same compartment, one needs to increase the resolution of the microscopy technique used.

In principle, nanobarcoding should be suitable for super-resolution analyses, especially as the probes used (nanobodies) have been heavily used in super-resolution for a decade (e.g., [36]). One limitation is that super-resolution imaging tools have been notoriously difficult to apply to more than 2 to 3 color channels, although improved hardware and spectral demixing algorithms may alleviate this problem [37]. A number of other strategies have also emerged, which could be employed for multichannel observations. First, one could rely fluorescence lifetime detection, to separate spectrally similar fluorophores [38], or, in a more advanced implementation, one could use single-molecule spectroscopy, for the same purpose [39]. An often used approach for multiplexing, as mentioned in the introduction, is DNA-PAINT, for which all our nanobodies are readily available, some have already been used for PAINT multiplexing [4]. In fact, nanobody-based PAINT barcoding is now being used to identify endogenous proteins in neurons, relying on primary antibodies bound by secondary nanobodies carrying different barcodes [40], albeit these procedures require extensive buffer exchanges and repeated imaging, something we aimed to avoid in our current approach.

To maintain the ease of use of multicolor imaging experiments, but obtain a high resolution, one could rely on expansion microscopy, in which the sample is labeled with nanobodies, exactly as we now performed, and is then embedded in a swellable gel and is expanded [41]. This type of procedure could raise the resolution of the images by at least 5- to 10-fold, depending on the expansion factor of the gel, without major changes to our overall approach. Multicolor images have been obtained with this approach, at very high resolutions [42,43].

Another potential limitation is the use of genetic encoding, since multiple constructs need to be introduced into the same cells. Current developments in CRISPR/Cas technologies should render this approach not overly difficult, as cell lines containing multiple constructs can be readily obtained. In addition, the sequences (barcodes) could be expressed, purified, and linked to secondary nanobodies, which are applied to reveal primary antibodies in immunocytochemistry and are inherently multiplexable, as explained above for DNA-PAINT (see also [44]), thereby extending the assay to many protein targets. Finally, since many other barcode epitopes could be used, our approach should have a large application range in the field of cellular biology and proteomics.

Our deep learning approach adds to a rapidly growing body of work in the imaging field. Similar deep learning methods, for example, image segmentation [45–48] and feature detection [49,50], are among the most sought-after applications in imaging. Other prominent applications include resolution enhancement [51–53] and increasing the signal-to-noise ratio [54]. Efforts are being made to democratize the application of deep learning in microscopy for

nonexperts through open-source solutions [55–58]. Given these developments, we assume that future implementations of nanobarcoding will become increasingly easier to analyze and, therefore, more easily applicable.

## Materials and methods

### In silico design of nanobarcode proteins for protein identification

Nanobarcode proteins were designed in silico and consist of 3 main components: (1) the protein sequence; (2) up to 4 genetically encoded epitopes that form the nanobarcode; and (3) the ALFA-tag [11] for testing purposes. The used epitopes have been validated previously and/or in this manuscript (for an overview, see S1 Table). Short flexible linkers (5 amino acids long) were added in between epitopes to ensure epitope availability. We used abbreviations for the nanobarcode proteins to ensure readability (Fig 1B). For example, the abbreviation NLS(1101) was used for the nanobarcode protein NLS_L2_mCherry(Y71L)_L3_GFP(Y66L) _L4_α_L5_syn2. It contains 3 of the 4 nanobarcode epitopes, mCherry(Y71L), GFP(Y66L), and syn2, plus the ALFA-tag for testing purposes. Accordingly, the NLS construct contains 4 flexible linkers (L2 to L5). The position of the ALFA-tag and the positions of the flexible linkers varied among the palette of proteins used, according to the characteristics of each protein. Full-length sequences are uploaded to the repository as an overview table and as single.ape files (http://dx.doi.org/10.17169/refubium-40101/Plasmid_design.zip). The company Gen-Script Biotech generated the pcDNA3.1(+) mammalian expression vectors containing the nanobarcode sequences DNA, using the NheI/XhoI cloning sites.

### Cell culture experiments with human embryonic kidney 293 cells (HEK293 cells)

**HEK293 cell cultures.** HEK293 cells were cultured in a $CO_2$ incubator (37°C and 5% $CO_2$). Cells were maintained in Dulbecco's Modified Eagle Medium (Sigma-Aldrich, 10% fetal bovine serum (Sigma-Aldrich), 2% L-Glutamine (Gibco), 0.6% penicillin/streptomycin (Lonza). When confluent, cells were briefly washed with DPBS (Dulbecco's Phosphate Buffered Saline) and were detached from the culture dishes using 0.05% Trypsin–EDTA (1X, Gibco), before culturing them in fresh medium. Approximately 24 hours before plasmid transfection, HEK293 cells were seeded on 12-well plates containing poly-L-lysine–coated glass coverslips. An exception was made for the co-seeding assay (see below), in which the cells were first mixed in solution, before finally being seeded on glass coverslips.

**Lipofectamine-based transfection of HEK293 cells.** HEK293 cells were transfected with 1 to 2 μg of plasmid DNA per well, after mixing with 1.5 to 4 μl of Lipofectamine 2000. The optimal time window for transfection was defined based on the protein performance in the neural network analysis (Figs 2C, 2D, S19 and S20); see below for details about the neural network identification procedure. Independent tested time windows were overnight ($N = 1$), 24 hours ($N = 2$), 48 hours ($N = 2$), and 72 hours ($N = 1$) for each protein tested.

**Co-seeding of cell suspensions containing distinct HEK293 cell populations.** For co-seeding, transfected HEK293 cells were trypsinized, washed, and brought to suspension in complete medium without antibiotics. Subsequently, cells transfected with different constructs were mixed. Up to 11 different transfected cell populations were co-seeded into a single 12-well plate (CellStar) containing poly-L-Lysine–coated coverslips. Plates with co-seeded cells were gently shaken in a humidified incubator for 1 hour and then stopped, allowing cells to attach to the coverslips overnight.

## Nanobody production and coupling

Nanobodies were custom produced (NbSyn87 and NbSyn2) or simply purchased as catalog products from NanoTag Biotechnologies GmbH (Göttingen, Germany), as described below.

## Immunocytochemistry procedure

**Immunocytochemistry with antibodies and or nanobodies.** Transfected HEK293 cells were fixed with 4% PFA for 45 minutes at room temperature, followed by a short rinse in PBS and aldehyde quenching with 100 mM $NH_4Cl$ and 100 mM glycine in PBS, for 30 minutes at room temperature. Cells were permeabilized and blocked using PBS supplemented with 0.01% Triton X-100 and 2% bovine serum albumin (BSA) for 30 minutes at room temperature. Permeabilized cells were immunostained with the following fluorescent nanobodies: NbSyn87 (conjugated to DyLight 405), NbEGFP (FluoTag-Q anti-GFP Atto488, Cat#N0301-At488-L), NbRFP (conjugated to Atto565, sold as FluoTag-Q anti-RFP, Cat#N0401-AT565-L), and NbSyn2 (conjugated to Star635P). All nanobodies have been characterized and used in early studies (the NbEGFP and NbRFP [4] and the NbSyn2 and NbSyn87 [13,59–61]). Nanobodies were incubated for 1 hour at room temperature in the permeabilization/blocking buffer indicated above, at final concentrations of approximately 70 nanogram/µl (NbEGFP), 70 nanogram/µl (NbRFP), approximately 70 nanomolar (NbSyn87), and approximately 70 nanomolar (NbSyn2). Excess nanobody was thoroughly washed with PBS, and coverslips were mounted on microscope slides using Mowiol. For antibody stainings (S2, S5, S16 and S21 Figs), procedures were very similar, but now, stainings were achieved by 1-hour incubation with primary antibodies followed by a 30- to 60-minute incubation with secondary antibodies (for details regarding antibodies and used concentrations, see S2 Table).

## EGF and transferrin assay

During EGF and transferrin pulse and chase, 12-well plates with HEK cells were kept in a water bath (37°C). As a first step, the cell medium was removed and washed twice with pre-warmed HBSS (minimal buffer, no calcium, no magnesium, no phenol red). HBSS was supplemented with 1 mM $CaCl_2$ and 1 mM $MgCl_2$ directly before use. Transferrin-Alexa488 (Thermo Fisher, E35351) and EGF-Alexa647 (Thermo Fisher, T13342) stock solutions were prepared and stored according to the company's instructions. For our experiments, stock solutions were diluted 1:100 in HBSS. HEK cells were pulsed with transferrin and EGF for 10 minutes, allowing the cells to endocytose these ligands. Afterwards, they were immediately fixed or were chased (washed off) in a minimal buffer at 37°C, for 10 or 20 minutes. Finally, all cells were fixed and immunolabeled for the ALFA tag (see immunocytochemistry procedure) to identify the nanobarcoded proteins.

## Imaging and image processing

Multichannel images were obtained using a ZEISS LSM 710 AxioObserver equipped with a ZEISS Plan Apochromat 63× oil DIC objective lens (NA 1.40). Images were acquired using $512 \times 512$ pixels, at 440 nm pixel sizes. Samples were illuminated with the following lasers (fiber launching): $\lambda = 405$ nm, $\lambda = 488$ nm, $\lambda = 561$, and $\lambda = 633$ nm, excitation filters MBS 488/561/633 and MBS 405. Fluorescence was collated using the corresponding diffraction grating and bandwidth slit settings for emission 416 to 485 nm (CH1), 494 to 554 nm (CH2), 572 to 632 nm (CH3), and 641 to 730 nm (CH4).

The following combinations of lasers (excitation) and bandwidth slit settings were used for our 10-channel recordings:

1. 405 nm plus CH1-4 (images 1 to 4)

2. 488 nm plus CH2-4 (images 6 to 8)

3. Brightfield image for validation purposes (image 5)

4. 561 nm plus CH3-4 (images 9 and 10)

5. 641 nm plus CH4 (image 11)

Validation images of the fluorescent protein constructs stained with primary and secondary antibodies (see S2 Fig and S2 Table) were obtained using an Olympus IX71 microscope equipped with an Olympus UPlanSApo 60× oil objective (1.35 NA). These validation images were only obtained to determine whether the proteins behaved as expected. Rab5a images of S2 Fig and all work relating to the deep neural network was performed on the ZEISS LSM 710 AxioObserver.

## Super-resolution assay

Transfected PC12 cells, a neuroblastoma cell line endogenous expressing SNAP25, were transfected with the SNAP25(1100) construct, fixed, and stained with anti-SNAP antibody and NbALFA. Transfection, fixation, and staining were similar to the procedures described above in the "Lipofectamine-based transfection of HEK293 cells" and "Immunocytochemistry with Antibodies and or Nanobodies" Methods sections. The only difference was that, this time, the stainings were done sequentially, starting with NbALFA (30 minutes) and followed by the primary and secondary antibodies against SNAP25 (for details about the antibody dilutions, see S2 Table).

## STED imaging

STED images were obtained using the Abberior Quad Scan Super-Resolution Microscope. Before imaging, a laser power meter (Thorlabs) was used to measure and set the energy levels of the lasers used (approximately 1 μW for excitation, approximately 5 mW for depletion). Images (pixel size: 50 nm; 1,400 × 1,400 pixels) of endogenous SNAP25 were obtained using the Abberior STAR580 excitation and emission settings (561 nm excitation/775 nm depletion) of the Imspector software. For images of SNAP25(1100), we used the Abberior STAR635P settings (640 nm/775 nm depletion). Dwell time was set to 10.00 μs and line averaging to 2.

## Data pipeline for training and evaluation of the deep network

The data for supervised training of the network are prepared by taking images of single transfects with known nanobarcode proteins. To separate the foreground (fluorescing nanobarcodes) from the background, we have used the kPCA algorithm [21]. The kPCA algorithm learns a nonlinear map with a prespecified kernel function that transforms the data such that maximum standard deviation is achieved along a reduced number of dimensions. For each case, which contains a nanobarcode with a time window (e.g., SNAP25, 48 hours), we train one kPCA model with a single reduced dimension over 4,000 pixels randomly selected from all the available images and then apply the trained model to all the pixels (Fig 3). We found out that applying a relative threshold at 0.8 of the range of the transformed values amounts to a reliable separation of pixels into foreground and background (Fig 3). We aimed to gather a maximum of 10,000 pixels per each case, but the actual available number could be smaller due to the quality of captured images (S3 Table). To each sample containing proteins, we gathered

10 blank samples to capture the background noise in the absence of any nanobarcodes (S3 Table).

For the training procedure to cover situations where the actual signal-to-noise ratio is lower than the gathered samples, we applied a contrast augmentation to the training data. We scale the values of channel intensities in the range 0.5 to 1.5 in a stochastic manner each time the network accesses the training data. The approach results in a more robust prediction as well as less chance of overfitting. The effect of this on-the-fly data augmentation can be traced in validation accuracies being higher than the training accuracies in the training loop of the deep network (Fig 3C).

## Deep neural network for nanobarcode identification

We have designed and trained a deep neural network for the identification of protein nanobarcodes from multichannel confocal images (S18 Fig). We have taken an approach similar to image segmentation and have assigned to each pixel of the image probabilities corresponding to the presence of each of the nanobarcodes, i.e., the network learns a mapping between channel intensities per pixel to a Multinoulli probability distribution. The components of the vector $\mathbf{x} = (x_1, \ldots, x_n)^T$ represent intensities of each of the $n$ imaging channels. This vector is fed to the network as the input, producing an $m$-dimensional output $\mathbf{y} = (y_1, \ldots, y_m)^T$. The components of the vector $\mathbf{y}$ are related to the probabilities $P(\mathbf{x} \in C_i; \boldsymbol{\theta})$, with $C_i$ denoting the fluorescence data pertaining to the $i$-th nanobarcode class, and $\boldsymbol{\theta}$ being the parameters of the network. In order for the output of the network to be a normalized probability, the last layer applies a softmax function (S18B Fig),

$$P(\mathbf{x} \in C_i; \boldsymbol{\theta}) = \mathrm{softmax}(y_i) = \frac{\exp(y_i)}{\sum_{j=1}^{m} \exp(y_j)}$$

The loss function, minimizing which with respect to $\boldsymbol{\theta}$ constitutes the training procedure, is the negative log-likelihood of the Multinoulli probability distribution,

$$\mathcal{L} = -\frac{1}{N} \sum_{n=1}^{N} \log(P(\mathbf{x} \in C_{i(n)}; \boldsymbol{\theta}))$$

where $N$ is the number of samples in one mini-batch, and $i(n)$ is the target class assigned to the $n$-th sample. Maximizing the log-likelihood over the probabilities is equivalent to minimizing the cross-entropy between the target distribution (which sharply separates classes in a one-hot representation) and the distribution modeled by the network [5].

We have designed the feed forward network by stacking residual blocks (S18 Fig). Using residual learning allows the training of significantly deeper network [62] In addition, by increasing cardinality, i.e., inclusion of multiple parallel paths through the network, we allow for high representation power with less network depth, thus preventing vanishing gradients during the training procedure [63] (S18C Fig). Dense (or fully connected) layers of the network apply an affine transformation to their input, followed by the nonlinear activation function $g$. Thus, for vectors $\mathbf{z}$ being transformed by the network at a dense layer, $\mathbf{z}_{\mathrm{out}} = g(\mathbf{W}\mathbf{z}_{\mathrm{in}} + \mathbf{b})$, where the weights matrix $\mathbf{W}$ and the bias vector $\mathbf{b}$ are trainable parameters of the layer. We have used Rectified Linear Unit (ReLU) as the activation function $g$ throughout the network and have employed the batch normalization algorithm to regularize the processed data during training and achieve better convergence [64] (S18 Fig).

## Training and testing the deep network

The network is trained via gradient descent using the AdamW algorithm [65,66]. Gradient of the loss function with respect to trainable parameters is calculated in the forward pass, and the optimizer algorithm updates the parameters via backpropagation [67]. We use a starting learning rate of $5 \times 10^{-4}$ for the AdamW optimizer and apply a step-decay of 0.9 per each 20 epochs. A batch size of 458 is used (see "Hyperparameter optimization" for details).

The input data are split 80%–10%–10% into training, validation, and hold-out test datasets, with the network being trained only on the training set, and the training procedure monitored via the loss and accuracies obtained with the validation set. We found out that training the network beyond 100 epochs is not necessary, as the validation loss plateaus before that, implying that the network might begin to overfit to the training data (Fig 3C). We applied early stopping by picking the trained network at an epoch after which the validation accuracies start to decline (Fig 3C).

After the training of the network is complete, the inference is done on full-sized images by feeding them to the network in a pixel-by-pixel scan. We have produced output images by assigning false colors to each protein and using output probabilities to compose a weighted color sum per pixel (S20 Fig). In order to account for slight variations in imaging conditions, we additionally applied trainable shifts and scales to the input channel intensity values via a contrast-modifier network. These transformations are separately trained per image in a self-supervised manner via minimizing the total entropy of the output signal. For each image, 50 steps of training are done with the contract-modifier. Apart from this, we have applied no other pre- or postprocessing to the images.

The performance of the trained network is evaluated based on the following metrics:

$$\text{accuracy} = \frac{\text{true pos.} + \text{true neg.}}{\text{all pos.} + \text{all neg.}}, \quad \text{precision} = \frac{\text{true pos.}}{\text{all pos.}}$$

$$\text{recall} = \frac{\text{true pos.}}{\text{true pos.} + \text{false neg.}}, \quad \text{F1} - \text{Score} = \frac{2}{\text{precision}^{-1} + \text{recall}^{-1}}$$

When the hold-out test set is used for evaluation, true/false positives and negatives are determined based on predicted and target classes. When the inference is done on images with a known nanobarcode, it is assumed that all the predictions not pertaining to blank or background should coincide with this nanobarcode. Thus, precision is the metric more suitable for this evaluation (S20 Fig).

## Hyperparameter optimization

Apart from trainable parameters, the network design contains a set of so-called hyperparameters, which in our case are the maximum layer width in each branch, number of branches, depth of each branch, batch size, and the learning rate. We have used the Adaptive Experimentation Platform (AX) to optimize hyperparameters based on the network performance on the validation set, i.e., its prediction accuracy after a fixed number of training epochs. The AX platform yields optimal values for the hyperparameters via Bayesian optimization [68].

## Supporting information

**S1 Fig. Design and topology of protein constructs.** (**A**-**C**) Legends for expected topology (**A**), protein length (**B**), and construct epitopes (**C**). (**D**) Protein topology schemes for the 15 constructs used. Below is a list with detailed information about the respective topology scheme of

each construct depicted in (**B**). Uniprot accession numbers (acc.nr.) are available under https://www.uniprot.org/uniprot/. Sequences of all constructs are listed in "plasmid_sequence_information.xlsx" stored in "Plasmid_design.zip" available from http://dx.doi.org/10.17169/refubium-40101. No protein, used for background signals.
(TIF)

**S2 Fig. A visualization of nanobarcode-carrying proteins using antibodies.** We validated the correct nanobarcoding and expression of the protein constructs by simultaneous visualization of the nanobarcodes and their respective endogenous epitope counterparts. The nanobarcodes were visualized by imaging their GFP or mCherry fluorescence (relying on constructs lacking the Y/L mutations of the chromophores), or by nanobody stainings, for barcodes lacking GFP or mCherry. We immunostained the respective proteins of interest with antibodies directed against protein-specific epitopes. (**A**) All protein constructs lacking mCherry or GFP fluorescence. (**B**, **C**) Protein constructs with GFP fluorescence. These proteins exhibit a strong localization to the perinuclear area, where antibodies penetrate more poorly than nanobodies [69]. See S3 and S4 Figs for nanobody staining of the nanobarcode epitopes. (**D**, **E**) All protein constructs having a fluorescent mCherry epitope. (**F**) All protein constructs having mCherry and GFP fluorescence. To visualize the target protein component of the nanobarcoded proteins, Cy5-coupled secondary antibodies were used. Scale bars: 10 μm.
(TIF)

**S3 Fig. Visualization of 15 nanobarcode epitopes using 4 spectrally distinct nanobodies.** (**A**-**D**) Nanobody-based identification of the 4 genetically encoded nanobarcode epitopes mCherry(Y71L), GFP(Y66L), syn87, and syn2 and the ALFA-tag epitope by their corresponding nanobodies NbRFP, NbEGFP, NbSyn87, NbSyn2, and NbALFA. Scaling was optimized for each protein. (**D**) VAMP4(1111) example (all epitopes present) and negative control condition: mock transfection (no DNA, no epitopes present) using same intensity scale. (**E**) As in (**D**), now with upscale intensities. Scale bar: 50 μm.
(TIFF)

**S4 Fig. Simultaneous nanobody staining of multiple epitopes.** (**A**) Nonfluorescent epitopes of SNAP25(1100) are recognized successfully by the corresponding nanobodies, independent of the number of nanobodies used. The SNAP25(1100) construct is successfully stained when using the NbALFA only (first row), when using the anti-GFP and anti-RFP nanobodies (second row) or when using the anti-GFP, anti-RFP, and anti-ALFA nanobodies (third row). The corresponding nanobarcode epitopes are detected, which indicates that there is no substantial steric hindrance between the nanobodies (which is expected due to the small size of the nanobodies). The morphological features of the cells are similar to cells transfected with a SNAP25 construct containing fluorescent epitopes (**B**), which indicates that these constructs are comparable and, therefore, suitable for our investigations, as shown in Figs 3 and 4. Scale bars: 30 μm.
(TIF)

**S5 Fig. Visualization of nanobarcoded proteins that act as markers for specific organelles.** The proteins indicated in the left-most column are markers for specific compartments, indicated in the next column. The colocalization of these proteins and specific compartment markers is then indicated in the fluorescence images. Scale bars: 20 μm. For quantification, see S17 Fig, below.
(TIF)

**S6 Fig. A functional assay to test nanobarcoded proteins.** Cells expressing different nanobarcoded proteins were pulsed with transferrin conjugated to Alexa488 and with EGF conjugated to Alexa647, for 10 minutes, allowing the cells to endocytose these ligands. Afterwards, they were immediately fixed or were chased (washed off) in a minimal buffer at 37˚C, for 10 or 20 minutes. Finally, all cells were fixed and immunolabeled for the ALFA tag, to identify the nanobarcoded proteins. (**A**, **B**) The behavior of transferrin and EGF, respectively. Transferrin recycles, as expected, being released during the chase period (Kruskal–Wallis test followed by Tukey post hoc test, $p < 0.006$ for endocytosis vs. 10- or 20-minute wash-off). No changes were seen for EGF, as expected (Kruskal–Wallis test, not significant). $N = 17$–18 independent experiments. (**C**, **D**) Same data as above, but indicating the nature of the nanobarcoded protein in each of the independent experiments. The data underlying this Figure can be found in the following Sheets of the "S1 Data file: "Tf_SFig 6A," "EGF_SFig 6B," "Tf_SFig 6C," and "EGF_SFig 6D." The S1 Data file is available from http://dx.doi.org/10.17169/refubium-40101. (TIF)

**S7 Fig. Transferrin and EGF imaging assays, tested for nanobarcoded Vti1a.** (**A**) Visualization of transferrin-Alexa488 (green) and EGF-Alexa647 (magenta), as well as the transfected protein, visualized with the ALFA nanobody (NbALFA) conjugated to AZdye568 (white). The 3 rows show the 10-minute pulse with the ligands (endocytosis), followed by the 10- and 20-minute chase (wash-off). To enable optimal visualization, the images are scaled differently, with the image scaling indicated in all panels. Scale bars: 20 µm. (**B**) The nanobarcoding scheme and the expected localization of the protein. (**C**) The NbALFA fluorescence intensity is plotted against the transferrin (green) and EGF (magenta) intensity, for all signals measured in 2 independent experiments, for all conditions. All intensities were normalized to the medians of the distributions and were then grouped in 20 bins of ALFA intensity, each containing similar numbers of values. The mean and SEM of each bin in the respective channels are plotted. The data underlying this Figure can be found in the S1 Data file, Sheet "SFig 7C_Vti1a," available from http://dx.doi.org/10.17169/refubium-40101. (**D**) The Pearson's correlation coefficients for the distributions from panel C are shown, with the *p*-values corrected for multiple testing using a Bonferroni correction. (TIF)

**S8 Fig. Transferrin and EGF imaging assays, tested for nanobarcoded syntaxin 4.** (**A**) Visualization of transferrin-Alexa488 (green) and EGF-Alexa647 (magenta), as well as the transfected protein, visualized with the ALFA nanobody (NbALFA) conjugated to AZdye568 (white). The 3 rows show the 10-minute pulse with the ligands (endocytosis), followed by the 10- and 20-minute chase (wash-off). To enable optimal visualization, the images are scaled differently, with the image scaling indicated in all panels. Scale bar: 20 µm. (**B**) The nanobarcoding scheme and the expected localization of the protein. (**C**) The NbALFA fluorescence intensity is plotted against the transferrin (green) and EGF (magenta) intensity, for all signals measured in 2 independent experiments, for all conditions. All intensities were normalized to the medians of the distributions and were then grouped in 20 bins of ALFA intensity, each containing similar numbers of values. The mean and SEM of each bin in the respective channels are plotted. The data underlying this Figure can be found in the S1 Data file, Sheet "SFig 8C_STX4," available from http://dx.doi.org/10.17169/refubium-40101. (**D**) The Pearson's correlation coefficients for the distributions from panel C are shown, with the *p*-values corrected for multiple testing using a Bonferroni correction. (PNG)

**S9 Fig. Transferrin and EGF imaging assays, tested for nanobarcoded syntaxin 6.** (**A**) Visualization of transferrin-Alexa488 (green) and EGF-Alexa647 (magenta), as well as the transfected protein, visualized with the ALFA nanobody (NbALFA) conjugated to AZdye568 (white). The 3 rows show the 10-minute pulse with the ligands (endocytosis), followed by the 10- and 20-minute chase (wash-off). To enable optimal visualization, the images are scaled differently, with the image scaling indicated in all panels. Scale bar: 20 μm. (**B**) The nanobarcoding scheme and the expected localization of the protein. (**C**) The NbALFA fluorescence intensity is plotted against the transferrin (green) and EGF (magenta) intensity, for all signals measured in 2 independent experiments, for all conditions. All intensities were normalized to the medians of the distributions and were then grouped in 20 bins of ALFA intensity, each containing similar numbers of values. The mean and SEM of each bin in the respective channels are plotted. The data underlying this Figure can be found in the S1 Data file, Sheet "SFig 9C_STX6," available from http://dx.doi.org/10.17169/refubium-40101. (**D**) The Pearson's correlation coefficients for the distributions from panel C are shown, with the *p*-values corrected for multiple testing using a Bonferroni correction.
(TIF)

**S10 Fig. Transferrin and EGF imaging assays, tested for nanobarcoded syntaxin 7.** (**A**) Visualization of transferrin-Alexa488 (green) and EGF-Alexa647 (magenta), as well as the transfected protein, visualized with the ALFA nanobody (NbALFA) conjugated to AZdye568 (white). The 3 rows show the 10-minute pulse with the ligands (endocytosis), followed by the 10- and 20-minute chase (wash-off). To enable optimal visualization, the images are scaled differently, with the image scaling indicated in all panels. Scale bar: 20 μm. (**B**) The nanobarcoding scheme and the expected localization of the protein. (**C**) The NbALFA fluorescence intensity is plotted against the transferrin (green) and EGF (magenta) intensity, for all signals measured in 2 independent experiments, for all conditions. All intensities were normalized to the medians of the distributions and were then grouped in 20 bins of ALFA intensity, each containing similar numbers of values. The mean and SEM of each bin in the respective channels are plotted. The data underlying this Figure can be found in the S1 Data file, Sheet "SFig 10C_STX7," available from http://dx.doi.org/10.17169/refubium-40101. (**D**) The Pearson's correlation coefficients for the distributions from panel C are shown, with the *p*-values corrected for multiple testing using a Bonferroni correction.
(TIF)

**S11 Fig. Transferrin and EGF imaging assays, tested for nanobarcoded endobrevin.** (**A**) Visualization of transferrin-Alexa488 (green) and EGF-Alexa647 (magenta), as well as the transfected protein, visualized with the ALFA nanobody (NbALFA) conjugated to AZdye568 (white). The 3 rows show the 10-minute pulse with the ligands (endocytosis), followed by the 10- and 20-minute chase (wash-off). To enable optimal visualization, the images are scaled differently, with the image scaling indicated in all panels. Scale bar: 20 μm. (**B**) The nanobarcoding scheme and the expected localization of the protein. (**C**) The NbALFA fluorescence intensity is plotted against the transferrin (green) and EGF (magenta) intensity, for all signals measured in 2 independent experiments, for all conditions. All intensities were normalized to the medians of the distributions and were then grouped in 20 bins of ALFA intensity, each containing similar numbers of values. The mean and SEM of each bin in the respective channels are plotted. The data underlying this Figure can be found in the S1 Data file, Sheet "SFig 11C_Endo," available from http://dx.doi.org/10.17169/refubium-40101. (**D**) The Pearson's correlation coefficients for the distributions from panel C are shown, with the *p*-values corrected for multiple testing using a Bonferroni correction.
(PNG)

**S12 Fig. Transferrin and EGF imaging assays, tested for nanobarcoded LifeAct.** (**A**) Visualization of transferrin-Alexa488 (green) and EGF-Alexa647 (magenta), as well as the transfected protein, visualized with the ALFA nanobody (NbALFA) conjugated to AZdye568 (white). The 3 rows show the 10-minute pulse with the ligands (endocytosis), followed by the 10- and 20-minute chase (wash-off). To enable optimal visualization, the images are scaled differently, with the image scaling indicated in all panels. Scale bar: 20 μm. (**B**) The nanobarcoding scheme and the expected localization of the protein. (**C**) The NbALFA fluorescence intensity is plotted against the transferrin (green) and EGF (magenta) intensity, for all signals measured in 2 independent experiments, for all conditions. All intensities were normalized to the medians of the distributions and were then grouped in 20 bins of ALFA intensity, each containing similar numbers of values. The mean and SEM of each bin in the respective channels are plotted. The data underlying this Figure can be found in the S1 Data file, Sheet "SFig 12C_LifeAct," available from http://dx.doi.org/10.17169/refubium-40101. (**D**) The Pearson's correlation coefficients for the distributions from panel C are shown, with the *p*-values corrected for multiple testing using a Bonferroni correction.
(TIF)

**S13 Fig. Transferrin and EGF imaging assays, tested for nanobarcoded Rab5a.** (**A**) Visualization of transferrin-Alexa488 (green) and EGF-Alexa647 (magenta), as well as the transfected protein, visualized with the ALFA nanobody (NbALFA) conjugated to AZdye568 (white). The 3 rows show the 10-minute pulse with the ligands (endocytosis), followed by the 10- and 20-minute chase (wash-off). To enable optimal visualization, the images are scaled differently, with the image scaling indicated in all panels. Scale bar: 20 μm. (**B**) The nanobarcoding scheme and the expected localization of the protein. (**C**) The NbALFA fluorescence intensity is plotted against the transferrin (green) and EGF (magenta) intensity, for all signals measured in 2 independent experiments, for all conditions. All intensities were normalized to the medians of the distributions and were then grouped in 20 bins of ALFA intensity, each containing similar numbers of values. The mean and SEM of each bin in the respective channels are plotted. The data underlying this Figure can be found in the S1 Data file, Sheet "SFig 13C_Rab5a," available from http://dx.doi.org/10.17169/refubium-40101. (**D**) The Pearson's correlation coefficients for the distributions from panel C are shown, with the *p*-values corrected for multiple testing using a Bonferroni correction.
(TIF)

**S14 Fig. Transferrin and EGF imaging assays, tested for nanobarcoded syntaxin 13.** (**A**) Visualization of transferrin-Alexa488 (green) and EGF-Alexa647 (magenta), as well as the transfected protein, visualized with the ALFA nanobody (NbALFA) conjugated to AZdye568 (white). The 3 rows show the 10-minute pulse with the ligands (endocytosis), followed by the 10- and 20-minute chase (wash-off). To enable optimal visualization, the images are scaled differently, with the image scaling indicated in all panels. Scale bar: 20 μm. (**B**) The nanobarcoding scheme and the expected localization of the protein. (**C**) The NbALFA fluorescence intensity is plotted against the transferrin (green) and EGF (magenta) intensity, for all signals measured in 2 independent experiments, for all conditions. All intensities were normalized to the medians of the distributions and were then grouped in 20 bins of ALFA intensity, each containing similar numbers of values. The mean and SEM of each bin in the respective channels are plotted. The data underlying this Figure can be found in the S1 Data file, Sheet "SFig 14C_STX13," available from http://dx.doi.org/10.17169/refubium-40101. (**D**) The Pearson's correlation coefficients for the distributions from panel C are shown, with the *p*-values

corrected for multiple testing using a Bonferroni correction.
(TIF)

**S15 Fig. Transferrin and EGF imaging assays, tested for nanobarcoded VAMP4.** (**A**) Visualization of transferrin-Alexa488 (green) and EGF-Alexa647 (magenta), as well as the transfected protein, visualized with the ALFA nanobody (NbALFA) conjugated to AZdye568 (white). The 3 rows show the 10-minute pulse with the ligands (endocytosis), followed by the 10- and 20-minute chase (wash-off). To enable optimal visualization, the images are scaled differently, with the image scaling indicated in all panels. Scale bar: 20 μm. (**B**) The nanobarcoding scheme and the expected localization of the protein. (**C**) The NbALFA fluorescence intensity is plotted against the transferrin (green) and EGF (magenta) intensity, for all signals measured in 2 independent experiments, for all conditions. All intensities were normalized to the medians of the distributions and were then grouped in 20 bins of ALFA intensity, each containing similar numbers of values. The mean and SEM of each bin in the respective channels are plotted. The data underlying this Figure can be found in the S1 Data file, Sheet "SFig 15C_VAMP4," available from http://dx.doi.org/10.17169/refubium-40101. (**D**) The Pearson's correlation coefficients for the distributions from panel C are shown, with the *p*-values corrected for multiple testing using a Bonferroni correction.
(TIF)

**S16 Fig. Endogenous SNAP25 and SNAP25(1100) have a similar cellular distribution within SNAP25(1100) transfected PC12 cells.** (**A**) Visualization of both endogenous SNAP25 and SNAP25(1100) using SNAP25 specific primary and secondary antibodies, plus NbALFA. (**B, C**) Negative control experiments, leaving out either primary antibodies (**B**) or NbALFA (**C**). (**D**) Imaging control, using a mixture of the same secondary antibody with 2 distinct fluorophores (targeting both endogenous SNAP25 and SNAP25(1100)), to provide a visual indication of the maximum expected colocalization. Bottom part of the figure: legend for used symbols and schemes. Scale bars: 2.5 μm. For quantification, see S17 Fig.
(TIF)

**S17 Fig. An analysis of the colocalization of epitope-tagged proteins to their expected compartments.** The images from S5 and S16 Figs were analyzed by measuring the Pearson's correlation coefficient in different image regions. The box plot indicates the respective values, compared to a control, consisting of similar measurements across the same regions in the protein-of-interest channel, and mirrored regions in the compartment channel. All proteins show a colocalization that is significantly above the control values (Kruskal–Wallis test followed by Tukey post hoc test, *p* < 0.006 for all proteins). The data underlying this Figure can be found in the S1 Data file, Sheet "SFig 17_all_loc_func," available from http://dx.doi.org/10.17169/refubium-40101.
(TIF)

**S18 Fig. Deep neural network for nanobarcode identification.** (**A**) Schematic representation of experimental protocol for obtaining multichannel images of HEK293 cells transfected with a single protein construct. HEK293 cells are seeded (1) and transfected with the necessary DNA plasmids. After an incubation of at least 14 hours, the HEK293 cells, now expressing the protein constructs, are fixed and stained with nanobodies (2). Multichannel images from the respective cells (3) are used for the training of a neuronal network. Wavelengths of excitation lasers used: λ = 405 nm, λ = 488 nm, λ = 561 nm, and λ = 633 nm. Emission channels used: 417–485 nm (CH1), 495–553 nm (CH2), 573–631 nm (CH3), and 641–729 nm (CH4). (**B**) Architecture of the deep network used for protein identification from channel intensity values pertaining to each pixel. For the dense layers, given numbers indicate input and output

dimensions. The network contains 4 parallel branches in the middle (2 are shown), the outputs of which are summed and processed by the final layers. The branches are composed of sequential residual blocks with skip connections bypassing triplets of layers, as shown in the expansion panel to the left (further details in Methods section "Deep neural network-based protein identification"). (**C**) The output probability distributions of the network are used to render false color images that contain information on the identified proteins in each pixel. Scale bars: 50 μm.
(TIF)

**S19 Fig. Deep network performance metrics for different protein expression times: Prediction accuracies (top panel titles), as well as precision, recall, and F1-Score (shown in the same order for each protein in the top panels), false positives and false negatives (bottom panels).** Data are shown for overnight, 24 hours and 72 hours protein expression, respectively. The data underlying this Figure are available as file "FigS19.xlsx" from http://dx.doi.org/10.17169/refubium-40101. The metrics for 48 hours are shown in Fig 2C and 2D.
(TIF)

**S20 Fig. Deep network analysis results.** (**A**) The prediction accuracy matrix of trained deep networks, estimated over all the images in the dataset. To increase the complexity of the training and testing procedure, we expressed each construct for different time periods, and we then trained and tested the deep networks with all of these different datasets. Each row corresponds to a separate network that has been trained solely on the given dataset. Columns are the average pixel-wise prediction accuracy, assuming that all the pixels picked by the network in an image should belong to the protein with which the cells have been transfected. The given accuracy values may include effects of misexpressed proteins, weak fluorescence signals, and imaging noise. (**B**) From left to right, first column: merged channels (405 nm/CH1, 488 nm/CH2, 561 nm/CH3, 633 nm/CH4), before being processed by the network. Second column: images produced by assigning false colors to bright pixels, assuming that all the proteins in the image exactly match the given nanobarcode. Third column: output of the deep network, with each pixel given the false color representing the protein picked by the network. Colors are scaled based on class probabilities (Fig 2). Fourth column: false color output of the network overlaid on the gray "cell halos" produced from the brightfield images. Brightfield images have been processed to remove noise and background gradients and to enhance the contrast. (**C**, **D**) As (**A**) and (**B**), for additional nanobarcode proteins. The data underlying this Figure are available as file "FigS20_AC.xlsx" from http://dx.doi.org/10.17169/refubium-40101.
(TIFF)

**S21 Fig. Validation of transfection and expression of protein constructs within HEK293 cells.** To enable an analysis of Nrxn and Nlgn pairing, we coexpressed different Nrxn and Nlgn constructs with specific nanobarcode proteins. This enables us to provide the different Nrxn- and Nlgn-containing cells with a recognizable identity, without having to modify the additional proteins themselves by nanobarcode tagging. However, this implies that we need to verify whether the majority of Nrxn- or Nlgn-expressing cells also express the respective nanobarcode proteins. (**A**, **B**) Nanobody staining with anti-ALFA-Atto488 reveals cells expressing protein constructs with nanobarcodes. Antibody staining with mouse-anti-HA and Cy3-anti-mouse reveals cells expressing NRXN or NL constructs with HA-tags. An overlay of both signals (anti-ALFA in green and anti-HA in magenta) indicates double-transfected cells, which make up the majority of all cells. $N = 2$ independent experiments for each protein combination. Scale bars: 50 μm.
(TIF)

**S1 Table. Protein tag validation in literature and/or in this manuscript.**
(DOCX)

**S2 Table. Information about antibodies used for target protein validation purposes.** The first 19 antibodies listed are primary antibodies. Antibodies 20–23 are secondary antibodies. See Methods section for further information about the staining procedures.
(DOCX)

**S3 Table. Summary of the image data used for training and testing the deep network.** Number of confocal images obtained for each protein (with the given nanobarcode) as well as number of pixels that have been sampled for the deep learning dataset. All images have the same dimensions of 512 × 512 pixels. With 72-hour samples, each image contains 5 slices in a z-stack. Network training, validation, and testing are done only based on the subsampled pixels (with the numbers given in the last column), while the precision matrices in S20 Fig are obtained on full-frame images.
(DOCX)

**S1 Data. Numerical data underlying graphs.** Names of individual sheets correspond to figure panels for which the numerical data is used.
(XLSX)

## Acknowledgments

The authors acknowledge Ms. Christina Zeising for helping with general laboratory procedures.

## Author Contributions

**Conceptualization:** Mohsen Sadeghi, Lucas Mittelmeier, F. Buket Basmanav, Felipe Opazo, Silvio O. Rizzoli.

**Data curation:** Mohsen Sadeghi, Cristian A. Bogaciu, Guobin Bao, Merle Hoff, Lucas Mittelmeier.

**Formal analysis:** Mohsen Sadeghi.

**Funding acquisition:** Mohsen Sadeghi, F. Buket Basmanav, Frank Noé, Silvio O. Rizzoli.

**Investigation:** Daniëlle de Jong-Bolm, Cristian A. Bogaciu, Guobin Bao, Merle Hoff, Lucas Mittelmeier, F. Buket Basmanav.

**Methodology:** Daniëlle de Jong-Bolm, Mohsen Sadeghi, Cristian A. Bogaciu, Guobin Bao, Gabriele Klaehn, Merle Hoff, Lucas Mittelmeier, F. Buket Basmanav, Felipe Opazo, Silvio O. Rizzoli.

**Resources:** Frank Noé, Silvio O. Rizzoli.

**Software:** Mohsen Sadeghi.

**Supervision:** Daniëlle de Jong-Bolm, F. Buket Basmanav, Felipe Opazo, Frank Noé, Silvio O. Rizzoli.

**Validation:** Daniëlle de Jong-Bolm, Mohsen Sadeghi, Cristian A. Bogaciu, Guobin Bao, Gabriele Klaehn, Merle Hoff, Lucas Mittelmeier, Silvio O. Rizzoli.

**Visualization:** Mohsen Sadeghi, Cristian A. Bogaciu, Guobin Bao, Gabriele Klaehn, Merle Hoff, Lucas Mittelmeier, Silvio O. Rizzoli.

**Writing – original draft:** Daniëlle de Jong-Bolm, Mohsen Sadeghi, Silvio O. Rizzoli.

**Writing – review & editing:** Daniëlle de Jong-Bolm, Mohsen Sadeghi, Guobin Bao, F. Buket Basmanav, Felipe Opazo, Frank Noé, Silvio O. Rizzoli.

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
