## [Editor Report · Decision Letter 0]

16 Sep 2022

Dear Dr de Jong-Bolm, 

Thank you for submitting your manuscript entitled "Protein nanobarcodes enable single-step multiplexed fluorescence imaging" for consideration as a Short Report by PLOS Biology.

Your manuscript has now been evaluated by the PLOS Biology editorial staff, as well as by an academic editor with relevant expertise, and I'm writing to let you know that we would like to send your submission out for external peer review.

IMPORTANT: Please change the article type to "Methods and Resources" when you upload the additional metadata (see next paragraph).

Once your full submission is complete, your paper will undergo a series of checks in preparation for peer review. After your manuscript has passed the checks it will be sent out for review. To provide the metadata for your submission, please Login to Editorial Manager (https://www.editorialmanager.com/pbiology) within two working days, i.e. by Sep 20 2022 11:59PM.

Kind regards,

Roli Roberts

Roland Roberts, PhD

Senior Editor

PLOS Biology

rroberts@plos.org

---

## [Decision Letter · Decision Letter 1]

13 Nov 2022

Dear Dr de Jong-Bolm,

Thank you for your patience while your manuscript "Protein nanobarcodes enable single-step multiplexed fluorescence imaging." was peer-reviewed at PLOS Biology as a Methods and Resources Article. I am handling your manuscript whilst my colleague Roland is away from the office. Please accept my apologies for the delays that you have experienced during the peer review process. Your manuscript has been evaluated by the PLOS Biology editors, an Academic Editor with relevant expertise, and by three independent reviewers.

The reviews are attached blow. As you can see, the reviewers are positive about the multiplexing barcode approach and think the method will be of interest to the field. However, they raise several overlapping concerns, including the need to perform additional validation experiments, such as demonstrating that the tags do not affect the function of the target proteins or to fully demonstrate the performance of the neural network. They also raise concerns with the overall strength of the reporting in the manuscript and ask that additional methodological information is provided. As noted by Reviewers #2 and #3, we ask that you please make the datasets and code underlying the neural network fully accessible.

Based on the reviews and following discussion with the Academic Editor, it is clear that a substantial amount of work would be required to meet the criteria for publication in PLOS Biology. However, given our and the reviewer interest in your study, we would be open to inviting a comprehensive revision of the study that thoroughly addresses all the reviewers' comments. Given the extent of revision that would be needed, we cannot make a decision about publication until we have seen the revised manuscript and your response to the reviewers' comments. Your revised manuscript would need to be seen by the reviewers again, but please note that we would not engage them unless their main concerns have been addressed. 

We appreciate that these requests represent a great deal of extra work, and we are willing to relax our standard revision time to allow you 6 months to revise your study. Please email us (plosbiology@plos.org) if you have any questions or concerns, or envision needing a (short) extension.

**IMPORTANT - SUBMITTING YOUR REVISION**

*Resubmission Checklist*

*Published Peer Review*

*PLOS Data Policy*

*Blot and Gel Data Policy*

Sincerely,

Richard

Richard Hodge, PhD

Associate Editor, PLOS Biology

rhodge@plos.org

On behalf of:

Roland Roberts, PhD

rroberts@plos.org

REVIEWS:

Reviewer #1: Jong-Bolm et al present a clever approach to multi-protein imaging via protein barcoding, potentially enabling the detection of more than 10 different proteins simultaneously in wide-field microscopy. Images are analysed using a classic autoencoder that identifies proteins with an accuracy of at least 80% (on the subset of proteins the authors tested). However, before recommending publication, the following major concerns should be addressed:

1- The introduction of five different epitopes might be expected to affect protein structure, function and organisation inside the cell. The authors refer to widefield images taken with the antibodies targeting the barcodes proteins which, in my opinion, do not confirm that the presence of all these epitopes on a target of interest do not change its structure, function and organisation. The authors are requested to validate their approach by carrying out functional assays as well as super-resolution/CryoEM imaging system on a randomly selected subset of proteins to proof that indeed the presence of these tags do not affect their targets. And, if the presence of the tags does affect the underlying proteins, what is the minimal number of tags that can be used for this approach to yield trustable result, does that minimum number of tags provide additional advantage over single colour/single protein imaging which allows up to 3/4 proteins to be simultaneously identified, and does reducing the number of tags reduce the accuracy (by how much) of the auto encoder in accurately identifying proteins. 

2- It is not clear how can a pixel-wise autoencoder classify proteins imaged in epi (or even confocal) fluorescence where the axial excitation depth is large enough that many (many) proteins might emit onto the same pixel. This approach cannot replace the widely-used RGB colour format where a single pixel is represented by different colours representing the ratio of different proteins imaged on that pixel. The authors need to comment on that limitation and to inform the reader what kind of readout they should expect from this approach.

3- It is claimed that this approach can be easily extended to super-resolution microscopy, however, this seems to be non-trivial. The approach relies on 3/4 (spectrally-distinct) colour imaging. Multi-colour imaging using dSTORM has been successfully demonstrated using a maximum of 2 spectrally-distinct colours (e.g. 561 nm and 647 nm) given that 405 nm and 488 nm are used for activating fluorophores. Multi-colour imaging using spectrally-overlapping fluorophores was shown using prisms and diffraction-gratings - however, these approaches are not compatible with nanocoding (given that they rely on multiple distinct colours). Nanobarcoding can be combined with DNA-PAINT/STORM(not dSTORM) based imaging, but can the authors comment how this can be done and the limitations accrued from the merge and what can this approach deliver more compared to spectral-demixing approaches described in (https://doi.org/10.1038/nmeth.3528 or https://doi.org/10.1101/2021.12.23.473862 or https://doi.org/10.1021%2Facs.nanolett.9b00508) 

4- The imaging setup is not adequately described. The type of dichroic, emission and excitation filters is not described. It is not described whether the excitation is launched by fibre or free space launching. This information is important for others to replicate the imaging conditions - specially given that the entire method relies on the relative intensities of imaged colours.

5- The training of the NN is not well described either. The number of batches, epochs, etc. are not described and the NN (raw and trained) are not open-sourced or packaged. This tool, if appropritately validated as in 1, would need to be accessible to biologists for them to be able to replicate. The authors would also need to mention the time taken to train and deploy the NN and the computational environment used to achieve these. 

Reviewer #2: The authors set out the potential for microscopy in performing highly multiplexed imaging to determine the spatial organisation of multiple markers at one time, yet only a handful of channels being practically tractable. They present an approach for solving this problem by using combinatorial epitope labelling. A series of constructs are generated with different assortments of epitope tags including the ALFAtag, GFP and mCherry which are then detected using fluorophore-conjugated nanobodies in fixed cells, termed 'nanobarcoding'. The samples are excited with each of 4 wavelengths in turn and the pixel intensity values are determined across the series of images at with each set of emission filters. The per-pixel intensity values are then used to train an neural network to identify which "nanobarcode" is present. I agree with the authors about the importance of the barrier and that combinatorial tagging is an attractive approach.

My main comment is that the manuscript as prepared isn't giving me all the information I need to evaluate the method in a clear enough manner. The text is overly abbreviated, including the abstract. To give an example, on page 4 the sentence "we verified the accuracy (Suppl. Fig 3-4)... appeared to function well" is to capture 5 pages of data on the foundation of the method. For a methods paper I am expecting to see the choice of controls clearly explained, performed and stated, such as for antibody specificity and whether the conjugated proteins are being accurately expressed and to what levels (e.g. an assessment by western blotting as well as the images). Sometimes supplementary figures are referred to in advance of the main figures (e.g Supp Figs 6 and 7, page 5) and even the supplementary figures are not giving me the levels of information and explanation I need to properly assess the approach. It is also not clear whether each user would need to train and create their own neural network or if there is analysis software the authors developed that can be straightforwardly implemented by potential users (probably not if it is not obvious, which limits the generality of the approach).

In terms of the science, the nanobarcoding tags are very large. I question the choice of GFP and mCherry rather than smaller epitope tags. For presenting a method I would expect to see this choice evaluated and compared with other possible choices. The biological example of Nrxns and Nlgns used to illustrate the technique is not sufficiently well explained for the wide readership of a methods paper. The false positive and false negative rates are substantial, typically >10% which significantly limits the utility of the approach in my view. The authors discuss that the approach relies on exogenous expression and is the tags are currently too big for endogenous labelling. My view is that the size of the nanobarcodes is a significant limitation such that this technique is not really widely applicable and is actually ends up quite a niche approach, even if it is well implemented. At the same time I can't properly tell if it is successful/well implemented because the manuscript seems to be making too many assumptions about what their readers can discern without explanation.

Reviewer #3: In this paper, D. Jong-Bolm et al. present a novel approach for multiplexed fluorescence microscopy. This involves the use of nanobodies combined with a deep learning analysis method. Using protein engineering, they could generate proteins containing 5 genetically encoded epitopes that are recognized by nanobodies. They designed a proof of principle study on 15 different proteins using 4 fluorescent tags. Confocal microscopy was performed with a 10 fluorescent channel recording scheme and a custom-based deep learning analysis was designed. The proposed method as the potential to have a strong impact in the field of multi-color imaging and could be combined with CRISPR/Cas gene editing technologies to extend its application to endogenous proteins. The validation experiments for the nanobody-based barcoding strategy are robust and convincing. While the analysis approach using deep learning is very promising, I think that essential validation steps are missing and should be included in the manuscript before publication. 

1. Training procedure of the neural network: in the Materials and Methods section, lines 248-251, the authors describe how the input data was split into training and hold-out test datasets. However, the training performance was monitored by comparing the performance on the training and testing dataset. From this description it appears to me that the testing set was used as a validation set and that no "independent test dataset" was used to report the performance. Therefore, we lack information on the generalization capacities of the proposed approach. The authors should show that the method can be trained on a dataset (split in test and validation sets) and evaluated on a completely independent testing dataset (never seen by the network during training and not used to monitor training performance during training). 

2. Datasets: A description of the dataset is lacking. This description should include at least: the number of images for each protein, the number of images for each time period (o/n, 24h, 48h, 72h), the size of the images (in microns and number of pixels), the size of training crops (if used). In the material and methods section, it should be specified if the reported performance (in Fig. 2) was obtained after training on the full dataset, or only on the dataset including the subset of proteins that are shown in Fig.2. It should also be mentioned if all images (for different expression time periods) were included in the training set. 

3. Prediction accuracy: The prediction accuracy of the network is provided only for a subset of proteins in Fig.2C&D, however it is not clear how it compares with the performance on the other proteins. A careful description of the performance on all 15 proteins should be included as well as an ablation study showing how reducing/increasing the number of detected protein targets affect the prediction accuracy. Results of the deep learning approach should be compared to established hyperspectral imaging baselines.

4. In supplementary Fig. 6, the Neural network analysis results are presented, and the accuracy seems very low (below 0.4) for multiple proteins and multiple training/testing combinations. The authors should discuss these results in the manuscript and compare the results to the one obtained with established hyperspectral imaging baselines. The approach used to generate the results in supplementary fig. 6 should be described in the materials and methods section. 

5. It is not clear to me why the validation images were taken on a different microscope and if those images were used to validate the deep learning approach. On line 86 they mention that the performance was analyzed on the hold-out test sets and on full images of single-transfected samples. Are those images of single-transfected samples the same as the validation images described at line 206? If so, the authors should specify how they have checked the generalizability of the approach on different microscopes. 

6. Data availability statement: In the manuscript the data availability statement states that "The data presented in this study are available on request to the corresponding author", while in the section Data Availability found in the table preceding the manuscript it is stated that all data are fully available without restriction. Considering that this manuscript presents results on image analysis with deep learning, it is essential that the authors provide a curated and open-source dataset as well as a fully documented and open-source code. 

7. Reproducibility of the approach: The information on the deep learning-based image analysis procedure is insufficient to reproduce the analysis presented in this manuscript. The author should provide access to a fully operational analysis code, including the default configurations with optimized hyperparameters and images to test the approach. A more detailed description of the optimized hyperparameters should be also provided in the method section. 

Overall, if the above-mentioned major comments are addressed by the authors, I would recommend this paper for publication given that the deep learning approach shows robust and generalizable results on an independent test set. I believe that this study can have a strong impact in the fields of multiplexed fluorescence microscopy and cell biology.

---

## [Decision Letter · Decision Letter 2]

20 Oct 2023

Dear Dr de Jong-Bolm,

Thank you for your patience while we considered your revised manuscript "Protein nanobarcodes enable single-step multiplexed fluorescence imaging." for publication as a Methods and Resources paper at PLOS Biology. This revised version of your manuscript has been evaluated by the PLOS Biology editors, the Academic Editor and two of the original reviewers.

Based on the reviews, we are likely to accept this manuscript for publication, provided you satisfactorily address the remaining points raised by the reviewers and the following data and other policy-related requests.

IMPORTANT - please attend to the following:

a) Please address the remaining requests from reviewer #3.

b) Please address my Data Policy requests below; specifically, we need you to supply the numerical values underlying Figs 2CD, 3ABCD, S6ABCD, S7C, S8C, S9C, S10C, S11C, S12C, S13C, S14C, S15C, S17, S19, S20AC, either as a supplementary data file or as a permanent DOI’d deposition. I note that you already have an associated GitHub deposition (https://github.com/noegroup/deep_nanobarcode); many thanks for doing this, however, because Github depositions can be readily changed or deleted, please make a permanent DOI’d copy (e.g. in Zenodo) and provide this URL (see below).

c) Please cite the location of the data clearly in all relevant main and supplementary Figure legends, e.g. “The data underlying this Figure can be found in S1 Data” or “The data underlying this Figure can be found in https://doi.org/10.5281/zenodo.XXXXX”

We expect to receive your revised manuscript within two weeks. 

*Published Peer Review History*

*Press*

Sincerely,

Roli Roberts

Roland Roberts, PhD

Senior Editor,

rroberts@plos.org,

PLOS Biology

DATA POLICY:

Regardless of the method selected, please ensure that you provide the individual numerical values that underlie the summary data displayed in the following figure panels as they are essential for readers to assess your analysis and to reproduce it: Figs 2CD, 3ABCD, S6ABCD, S7C, S8C, S9C, S10C, S11C, S12C, S13C, S14C, S15C, S17, S19, S20AC. NOTE: the numerical data provided should include all replicates AND the way in which the plotted mean and errors were derived (it should not present only the mean/average values).

CODE POLICY

Per journal policy, as the code that you have generated is important to support the conclusions of your manuscript, we require that you make it available without restrictions upon publication. Please ensure that the code is sufficiently well documented and reusable, and that your Data Statement in the Editorial Manager submission system accurately describes where your code can be found.

DATA NOT SHOWN?

REVIEWERS' COMMENTS:

Reviewer #1:

I have read the authors' responses and the revised manuscript with a lot of interest. I am particularly impressed with the authors' thorough implementation of functional assays to test the effect of their nanobarcodes on the different proteins they image. I would like to thank the reviewers for their extensive work on the revised manuscript - which must have been extensive - but, I can attest now that the revised manuscript is much better poised for publication and that their technology will have a profound impact in the field.

Reviewer #3:

The authors have carried out an in-depth revision of the manuscript. They have addressed most of the reviewers' comments and have greatly improved the description of the analysis method and the accessibility of their code. The Github repository where the code is shared is well documented. The notebooks are easy to use and run smoothly when following the set-up described on Github. The code is well documented, commented, and clearly partitioned into functions. The functions and their parameters correspond to what is described in the manuscript and the methods section.

Major comment : 

The effect of the spatial resolution on the reliability of the nanobarcoding approach and deep learning based analysis should be better demonstrated and described in the result section. A very short paragraph describing the aspect of having only one nanobarcode per pixel was included in the discussion but I think that this should be described with more details already with the presented results. For example, in the paragraph describing the results about the Nrxn and Nlgn interactions, it is not clear to me how pairs are identified and what is considered as being an interaction. If I understand it correctly, only a single nanobarcode can be associated to one pixel, so that the spatial resolution of the microscope and the distance between interacting proteins will influence the precision of the analysis approach. Was this controlled for the Nrxn and Nlgn pairs? How would the reliability of the technique be influenced by interacting proteins that are closer that the resolution limit of the chosen microscope? 

Notebooks

Both notebooks fail to load the network weights on a CPU machine. I suggest adding an 'if torch.cuda.is_available' statement when loading the weights. When False, the torch.load function should have the argument 'map_location=torch.device('cpu')'. 

When using the environment as provided, I get the error message "Error displaying widget" when running the last cells of both notebooks. Make sure the environment includes the libraries for displaying the widgets, if the widgets are necessary.

Minor comments: 

In Fig. 2a, it would be more helpful to have a less simplified version of the designed neural network, highlighting the principal structural elements of the network. 

Fig. 2b in the merged input column, the contrast is very low, making it very difficult to compare the signal with the NN output. 

Fig. 3b : please describe the red boxes in the legend and make sure that they are useful (it is not clear to me with this version of the figure why they have been added to the right panel). 

Fig. 3c : define the green and magenta colors in the legend

Lines 99-100 The approach appeared to function well, as illustrated by the images in Fig 1F, in which the nanobarcodes can be easily differentiated by the human observer. I suggest rephrasing : As illustrated in Fig 1F, the nanobarcodes can be easily differentiated by the human observer.

At line 167, authors should state why a trainable contrast modifier was needed and add a reference to support the use of such an approach in this context. Similarly, it is not clear on line 177 why 50 iterations of self-supervised contrast adaptation is required.

Lines 194-195, please add references for the listed methods. 

Lines 198-203 : Performing the ablation study was very helpful to investigate the class-dependent performance of the network. However, I would suggest rephrasing the paragraph between lines 198-203 as reducing the number of classes in an ablation study has also an impact on the task complexity, which can positively impact the overall performance. 

Line 206 : please define what is satisfactory precision. Was there a minimal precision that was identified in order for this method to be useful in the described experimental context? 

The added paragraph between lines 206 and 234 is difficult to follow as links between the sentences and with the main subject of the paper are often missing or unclear. I suggest reworking on this section of the paper to improve clarity. 

Line 1074 : cite the publication you are referring to in the statement The Nrxn/Nlgn codes, such as SS#4(+) refer to the respective splicing sites of the proteins, according to the literature.

I suggest rewording the following sentences to improve clarity and readability:

Line 137 : The enhancements can only be limited, since the membrane trafficking pathways remain limited by the abundance of many other proteins, which are not overexpressed. We obtained this result for all proteins. Suppl. Fig 6 presents an overall view of the results, indicating the transferrin and EGF dynamics in all experiments, combined.

Line 1020 (caption of figure 1) : Barcode epitopes recognized by fluorescent nanobodies are referred to as nanobarcodes and shown as "ones" in pseudo-colors that correspond to the fluorophores used. I suggest removing the added "are referred to as nanobarcodes"; from my understanding, the barcode epitopes (which are shown as "ones") are not referred to as nanobarcodes, it is the combination of four epitopes ("zeroes" and "ones") that are referred to as nanobarcodes.

Typos : 

Line 54 : data sets -> datasets

Line 132 : (whose binding to actin should lead to a small, but measurable enhancement of actin dynamics, [17], Remove the comma and replace it with a parenthesis.

Line 146 : (just as lowering SNAP25 levels in heterozygous SNAP25+/- mice leads to very minor phenotypes, [19]. Remove the comma and replace it with a parenthesis.

Line 164 : Labelled -> labeled

Line 168 : and is trained self-supervised -> and is trained in a self-supervised manner

Line 176 : utilizing the same hardware, the inference, takes about 15 seconds -> utilizing the same hardware, the inference takes about 15 seconds (remove the extra comma)

Line 219 : (e.g. [22] but has only recently been related to the synapse [23]. Close the parenthesis.

Line 460 : A batch size of 458 is uses. Change uses to used.

---

## [Editor Report · Decision Letter 3]

13 Nov 2023

Dear Dr de Jong-Bolm,

Thank you for the submission of your revised Methods and Resources article "Protein nanobarcodes enable single-step multiplexed fluorescence imaging." for publication in PLOS Biology. On behalf of my colleagues and the Academic Editor, Emma Rawlins, I'm pleased to say that we can in principle accept your manuscript for publication, provided you address any remaining formatting and reporting issues. These will be detailed in an email you should receive within 2-3 business days from our colleagues in the journal operations team; no action is required from you until then. Please note that we will not be able to formally accept your manuscript and schedule it for publication until you have completed any requested changes.

Sincerely, 

Roli Roberts

Senior Editor

PLOS Biology

rroberts@plos.org